# eDNA Increases the Detectability of Ranavirus Infection in an Alpine Amphibian Population

**DOI:** 10.3390/v11060526

**Published:** 2019-06-06

**Authors:** Claude Miaud, Véronique Arnal, Marie Poulain, Alice Valentini, Tony Dejean

**Affiliations:** 1CEFE, EPHE-PSL, CNRS, Univ. Montpellier, Univ Paul Valéry Montpellier 3, IRD, Biogeography and Vertebrate Ecology, 1919 route de Mende, 34293 Montpellier, France; veronique.arnal@cefe.cnrs.fr (V.A.); marie.poulain@cefe.cnrs.fr (M.P.); 2SPYGEN, 17 Rue du Lac Saint-André, 73370 Le Bourget-du-Lac, France; alice.valentini@spygen.com (A.V.); tony.dejean@spygen.com (T.D.)

**Keywords:** eDNA, *Ranavirus*, Common frog, *Rana temporaria*, early detection, virus surveillance

## Abstract

The early detection and identification of pathogenic microorganisms is essential in order to deploy appropriate mitigation measures. Viruses in the Iridoviridae family, such as those in the *Ranavirus* genus, can infect amphibian species without resulting in mortality or clinical signs, and they can also infect other hosts than amphibian species. Diagnostic techniques allowing the detection of the pathogen outside the period of host die-off would thus be of particular use. In this study, we tested a method using environmental DNA (eDNA) on a population of common frogs (*Rana temporaria*) known to be affected by a *Ranavirus* in the southern Alps in France. In six sampling sessions between June and September (the species’ activity period), we collected tissue samples from dead and live frogs (adults and tadpoles), as well as insects (aquatic and terrestrial), sediment, and water. At the beginning of the breeding season in June, one adult was found dead; at the end of July, a mass mortality of tadpoles was observed. The viral DNA was detected in both adults and tadpoles (dead or alive) and in water samples, but it was not detected in insects or sediment. In live frog specimens, the virus was detected from June to September and in water samples from August to September. Dead tadpoles that tested positive for *Ranavirus* were observed only on one date (at the end of July). Our results indicate that eDNA can be an effective alternative to tissue/specimen sampling and can detect *Ranavirus* presence outside die-offs. Another advantage is that the collection of water samples can be performed by most field technicians. This study confirms that the use of eDNA can increase the performance and accuracy of wildlife health status monitoring and thus contribute to more effective surveillance programs.

## 1. Introduction

Amphibians are often considered ecosystem health indicators (“the canaries in the coal mine”, [1]) due to their permeable skin, their sensitivity to environmental disturbance, and their often biphasic (requiring water and land) life cycle (e.g., [2]). Over 40% of the more than 7000 species that have been evaluated are classified as threatened today, and the main cause of the disappearance or decline of amphibian populations is the degradation and destruction of their habitats [3]. Emerging infectious diseases (EIDs) are also causing drastic amphibian declines worldwide [4,5,6], and the chytridiomycosis panzootic has been involved in the decline of at least 501 amphibian species over the past half-century, including 90 presumed extinctions [7]. *Ranaviruses* (family Iridoviridae)—double-stranded DNA viruses that infect fish, reptiles, and amphibians [8]—are considered the second most common infectious cause of mortality in amphibians [9,10]. These viruses have caused amphibian die-offs on five continents [11]. The common midwife toad virus (CMTV) [12] is a *Ranavirus* that has been causing amphibian die-offs since 2005 in Spain [10]. Since then, CMTV-related mortality has been recorded in other European countries, affecting several amphibian species in both Anura and Caudata [13,14].

The rapid and accurate identification of pathogenic microorganisms is essential for the early detection of infection and the deployment of appropriate mitigation measures [15]. Several field studies have reported *Ranavirus*-infected amphibians that do not present clinical signs or histological changes (e.g., [16]). The use of molecular techniques to identify microbial pathogens (e.g., see reviews in [17] for fungus and [18] for viruses) has considerably improved detection possibilities. Molecular tools applied to environmental samples are now widely used to identify infectious agents [19] and are particularly promising for the early detection of aquatic pathogens that can be introduced by non-native species (e.g., [20]). In the last decade, the DNA detection of pathogens in environmental samples (e.g., in water) has been performed successfully to identify several metazoan parasites, fungi (Chytrid and Oomycota), and *Ranaviruses* [21,22,23,24,25,26,27,28]. By comparing water samples and the *Ranavirus* infection status of wood frog tadpoles (*Lithobates sylvaticus*) in several ponds in north-eastern Connecticut in the United States, Hall et al. (2016) [24] demonstrated a strong relationship between the viral load in environmental DNA (eDNA) and larval tissues, indicating the effectiveness of eDNA-based *Ranavirus* detection in the field.

In Europe, *Ranavirus* infections and die-offs have been described in four amphibian families and ten species [8,13]. One of these species is the common frog (*Rana temporaria*), which is experiencing population die-offs in alpine lakes [13]. This study had two key aims: (1) To describe the infection status of different potential hosts of *Ranavirus* (common frog adults, tadpoles, and insects) and ecosystem compartments (sediment and water) in an alpine lake during the activity period (summer), both prior to and after an observed die-off; and (2) to use the eDNA method to detect *Ranavirus* during this infection event. We also compared previous studies using eDNA to detect Chytrids and *Ranavirus* to provide some recommendations for a more effective implementation of water sampling in monitoring programs.

## 2. Materials and Methods

### 2.1. Study Area

The study was conducted in Mercantour National Park in the south-eastern Alps in France. The sampled area consisted of several small lakes and a pond. Balaour pond (44.1082 N, 7.3742 E) is 25 × 20 m, and 2355 m a.s.l. It lies at a distance approximately 100 m from the closest lake. The maximum depth is 1.5 m, and the bottom is granitic rock, partly covered by a shallow (max 0.10 m) sedimentary layer of mud. There is no macrophytic vegetation.

The common frog *R. temporaria* uses these kinds of water bodies for breeding and hibernation in this alpine region [29]. In the study area, the common frog is the only amphibian species present, and it breeds only in Balaour pond (no breeding in the neighboring lakes). Many lakes are stocked with brown trout (*Salmo trutta*), and the common minnow (*Phoxinus phoxinus*), which is used as bait, has also been introduced in the lakes. However, there are no trout or minnows in the studied pond.

### 2.2. Water Temperature

The water temperature during the study was recorded using two data loggers (model iBCod 21G, Maxim/Dallas Semiconductor Inc., San Jose, CA, USA, accuracy +/− 1 °C). These were attached to a PVC tube (50 mm diameter; 40 cm in length) with one end glued to a 20 × 20 cm polystyrene float. One logger was positioned just below the float, and the other was positioned 25 cm from the first logger. They were programmed to record the temperature every two hours. The device was set in the field at 14:00 on the first day of the survey (11 June 2016) and remained there until 10:00 on the last day (19 September 2016). To do this, a 2 m high metallic pole was staked into the ground in the middle of Balaour pond (depth about 1 m). The PVC tube was then fitted to the pole with the float at the top. The data loggers were at a depth of about 5 and 30 cm, respectively. The PVC tube and float were designed to follow changes in water level so the data loggers recorded temperatures at a constant depth.

### 2.3. Amphibian Sampling

Sampling was carried out at Balaour pond on six dates between June and September in 2016 (Table 1), covering the developmental period of embryos and tadpoles in this region and at this altitude.

The pond was visually scanned to detect tadpoles and adults resting on the bottom. Live tadpoles were caught by a dip-net (mesh 1 mm). They were stored in individual plastic bottles (1.5 mL) filled with 95% ethanol. Dead tadpoles were collected and stored similarly. Live adults were caught by a dip-net (5 mm mesh), and the distal phalange of the second forelimb toe was collected and stored individually in a snap-cap tube (1.5 mL) filled with 95% ethanol. Disposable gloves were used and changed between each sampling, and the material (scissors and pliers) was soaked in ethanol and passed through a flame between each sample collection. The dead adults were collected with the dip-net and stored in a plastic bottle (300 mL) filled with 95% ethanol. Table 1 shows the sample size and developmental stages [30] of the tadpoles and adults collected. The toe clipping allowed us to check that unique adult specimens were sampled during the study period.

### 2.4. Insect Sampling

Both terrestrial and aquatic insects were collected in the six sampling sessions (Table 1). Terrestrial insects were collected using three yellow plastic plates placed on the ground about 10 m from the edge of the pond. These plates were filled with water and one drop of domestic detergent. After one hour of sampling, the flying insects found dead in the liquid were collected and stored in a 15 mL plastic bottle filled with 95% ethanol. Aquatic insects were caught by dip-netting (1 mm mesh), mostly in the shallow area of the pond (less than 0.60 m deep). To limit contamination with the surrounding water, each specimen was removed with needle-nose pliers and washed with distilled water. The aquatic insects collected on the same date were stored in the same 15 mL plastic bottle filled with 95% ethanol.

### 2.5. Water Sampling

The field survey method was modified from that used in [31]. Using a sterile water-sample dipper, a 100 mL water sample was collected at 20 locations equally spaced around the edge of Balaour pond, resulting in a pooled sample of approximately 2 L contained in a sterile self-supporting plastic bag. Samples were collected from the top 0.10 m of the water column preceded by a gentle circular movement with the sampling ladle. Surveyors stood on the pond bank without entering the water to avoid possible contamination from their boots or from stirring up sediment. The 2 L water sample was homogenized by gently shaking the bag to ensure the eDNA was evenly mixed throughout the sample, and the whole 2 L water sample was then filtered directly in the field through a VigiDNA 0.45 μm filter (SPYGEN, Le Bourget du Lac, France) using a sterile 100 mL syringe. The filter was filled with 80 mL of a CL1 conservative buffer (SPYGEN) and stored at room temperature before DNA extraction.

### 2.6. Sediment Sampling

At each of the six sampling sessions, sediment (approximately 10 mL) was collected from 20 locations evenly distributed around Balaour pond. Each sediment sample was collected approximately 0.5 m from the shoreline using a sterile syringe (20 mL) pushed into the top 5 cm of sediment. The 20 samples were placed into a shared sterile bag, mixed together, and then stored in a sterile wide-neck barrel (1 L).

### 2.7. Ranavirus Detection in Frogs, Tadpoles and Insects

From the adult frogs, a small piece (2 mm^3^) of tissue was collected: Liver tissue from dead specimens (which were dissected) and toe tissue from live specimens. For small tadpoles, the total tadpole was used (e.g., individuals <10 mm, Gosner stage 25). For larger tadpoles (Gosner stages 30–45), individuals were dissected, and a total tissue volume of about 2 mm^3^ was collected, mostly composed of the liver and heart. For both aquatic and terrestrial insects, specimens were grouped per date of sampling and crushed together in 5 mL of 95% ethanol. A subsample of 1.5 mL was collected for DNA analysis.

The DNA was extracted with a REDExtract-N-Amp Plant Kit (Sigma-Aldrich-Merck, Darmstadt, Germany) by incubating a piece of tissue or the bulk of insects in 50 µL of extraction solution at 95 °C for 20 min after ethanol evaporation (56 °C for 30 min). An equal volume of dilution solution was added to the extract to neutralize inhibitory substances before a polymerase chain reaction (PCR).

The Taqman real-time quantitative PCRs were performed following the protocol described in detail in [13], using primers and probes designed by Leung et al. (2017) [32]. All assays were performed in triplicate. *Ranavirus* DNA, provided in several densities by Stephen Price (Zoological Society, London, UK), was used to calibrate the standard curve.

A sample was considered positive if the amplification curves were similar to those in positive controls (i.e., shape, cycle threshold, values, and > 0.1 genomic equivalent), and at least two replicates gave positive amplification. If only one well resulted in a positive signal, the sample was rerun and was considered positive if at least two wells gave a positive amplification signal. The results were expressed in terms of genomic equivalent (GE) and prevalence (number of positive samples/total number of samples).

### 2.8. Ranavirus Detection in Sediment

The sediment samples were weighed, and a similar weight of a saturated phosphate buffer (Na_2_HPO_4_; 0.12 m; pH ≈ 8) was added, as described in [33]. The DNA extraction was performed using a commercial kit for soil DNA (NucleoSpin^®^ Soil; Macherey-Nagel, Düren, Germany), following the manufacturer’s instructions. The PCR amplifications were performed following the protocol previously described for amphibian tissue and insects.

### 2.9. Ranavirus Detection in Water

For the water samples, DNA extraction was performed following the protocol described in [34] in a dedicated room for water DNA extraction equipped with positive air pressure, UV treatment, and frequent air renewal. Before entering this extraction room, personnel used a connecting zone to change into full protective clothing, comprising a disposable body suit with hood, mask, laboratory shoes, overshoes, and gloves. All workbenches were decontaminated with commercial bleach, diluted to achieve a 0.5% sodium hypochlorite solution, before and after each manipulation. For DNA extraction, each filtration capsule, containing the CL1 buffer, was agitated for 15 min on an S50 shaker (cat Ingenieurbüro™) at 800 rpm, and then the buffer was emptied into a 50 mL tube before being centrifuged for 15 min at 15,000× *g*. The supernatant was removed with a sterile pipette, leaving 15 mL of liquid at the bottom of the tube. Subsequently, 33 mL of ethanol and 1.5 mL of 3 M sodium acetate were added to each 50 mL tube and stored for at least one night at −20 °C. The tubes were centrifuged at 15,000× *g* for 15 min at 6 °C, and the supernatants were discarded. After this step, 720 µL of an ATL buffer from the DNeasy Blood & Tissue Extraction Kit (Qiagen, Hilden, Germany) was added. The tubes were then vortexed, and the supernatants were transferred to 2 mL tubes containing 20 µL of Proteinase K. The tubes were then incubated at 56 °C for two hours. Subsequently, DNA extraction was performed using NucleoSpin® Soil (MACHEREY-NAGEL GmbH & Co., Düren, Germany) starting from step 6 and following the manufacturer’s instructions. The elution was performed by adding 100 µL of a SE buffer twice. After DNA extraction, the samples were tested for inhibition using real-time amplification following the protocol described in Biggs et al. (2015) [31], which involved adding a synthetic DNA sequence to each sample and then trying to amplify it. None of the samples were found to be inhibited. The samples were amplified using primers and probes designed by Leung et al. (2017) [32]. The qPCR was carried out in 12 replicates on a final volume of 25 µL, using 3 μL of template DNA, 12.5 μL of TaqMan Environmental Master Mix 2.0 (Life Technologies, Carlsbad, CA, USA), 6.5 μL of ddH2O, 1 μL of forward primer and reverse primer, and 1 μL of probe (MCP_probe) using thermal cycling at 50 °C for 5 min and 95 °C for 10 min, followed by 50 cycles at 95 °C for 30 s and 60 °C for 1 min. To detect potential contamination, qPCR negative controls and DNA extraction controls (with 12 replicates) were amplified in parallel. Standard curve calculations were based on three standards (4.5 × 10^7^, 4.5 × 10^4^, and 4.5 × 10^1^ target copies per 3 μL) made from a plasmid containing the viral MCP target. Samples were run on a BIO-RAD CFX96 Touch real-time PCR detection system in a room dedicated to amplified DNA analysis with negative air pressure and physically separated from the DNA extraction room.

The results were expressed in terms of the number of positive replicates/total number of replicates per sample and the mean number of DNA copies (using the number of copies per Rv+ replicates) per sample.

### 2.10. Ranavirus Identification

Viral DNA was obtained from 1 dead adult, 3 live adults, 5 dead tadpoles, and 5 live tadpoles. These 14 samples were sequenced following the Mao et al. (1999) PCR method [35], using the BigDye Terminator Cycle Sequencing kit (PE Biosystems, Thermo Fisher Scientific, Bleiswijk, Netherlands) on an ABI Prism 3100 (Applied Biosystems, Foster City, CA, USA). The electropherogram was exported and converted to Kodon (Applied Maths, Sint-Martens-Latem, Belgium), and, using the BLAST program (default settings), the sequences were compared to *Ranavirus* sequences previously identified in this region [15].

## 3. Results

### 3.1. Field Observations

At altitudes above 2000 m, the common frog breeds as soon as a small open body of water is available. The breeding population in Balaour pond has long been observed, but the population size has not been precisely estimated. The number of egg-masses is regularly around several hundred (M.-F. Lecchia, pers.comm.).

At Sampling Session 1 (10–11 June 2016), the breeding season had started, and about 200 egg-masses were counted. The embryos were at stages 9–11. Sixteen adults were observed in the water. One dead frog was observed at the bottom of the pond. At Session 2 (23–24 June), the tadpoles (stages 24 and 25) formed large aggregates on the remains of the spawn (jelly) along the pond shore. Fewer than 10 adults were observed in the water. At Session 3 (8 July), the tadpoles (stage 30) were grouped along the pond shore where the water temperature was highest. Five adults were observed in the water. At Session 4 (27 July), the tadpoles (stages 35–39) were grouped along the pond shore. Dead tadpoles (about 250 counted) were observed on the bottom of the pond. Several individuals were observed with atypical behaviour, such as slow movements when stimulated, lateral swimming, or lying on their back on the pond bottom (about 50 counted). Most of the dead tadpoles were eaten by their congeners (Figure 1). Only two adults were observed in the water. At Session 5 (15–16 August), the tadpoles (stages 39–41) were more widespread in the pond, i.e., observed swimming beyond the 2 m strip near the shoreline. While we did not estimate the (relative) density at each session, the number of live tadpoles observed at Session 5 was clearly lower than at Session 4. No dead tadpoles were observed, nor tadpoles with the previously described atypical behaviour. Only two adults were observed in the pond. At Session 6 (19 September), most of the tadpoles had metamorphosed or were very close to metamorphosis (stages 43–45) and were widespread along the pond shore, both in the water and on land. No dead tadpoles or froglets were observed. Eight adults were observed in the pond.

Terrestrial insects (Diptera, Chironomids, Hymenoptera, Rhopalocera, and ants) and aquatic insects (Diptera, Heteroptera, and Odonata) were present around and in Balaour pond from June to September.

The two data loggers allowed the water temperature to be recorded at the surface and at a depth of 30 cm (Figure 2). The temperature increased from about 7 °C in mid-June to 14 °C at the beginning of July. The mean temperature during the July–August sampling period was 14 °C ± 4.3, with a maximum of 20.5 °C (at the surface) and 20 °C (at a depth of 30 cm) in July. In September, the temperature decreased to about 7 °C.

### 3.2. Ranavirus Detection in Organisms

*Ranavirus* was detected in several compartments of the studied ecosystem (Appendix A and Figure 2) during the common frog activity period (June–September).

At Sampling Session 1 (10–11 June), the adult frogs present in the pond tested negative for *Ranavirus* (hereafter Rv-). The dead frog found on the pond bottom tested positive for *Ranavirus* (hereafter Rv+) with a genomic equivalent (GE) load of 2.83 × 10^1^ (Appendix A and Figure 2). At Session 2, 12 days later (23–24 June), three out of five adults were Rv+, with a mean GE of 0.43 ±. 1.45 (*n* = 3). Adults caught in the water continued to be Rv+ (four out of five testing positive) at Session 3 (8 July), with the highest mean GE (345 ± 14.5, *n* = 4, Figure 2). No adults were sampled at Session 4 (27 July). The individual caught at Session 5 (15–16 August) and the five caught at Session 6 (19 September) were all Rv- (Figure 2). Most of the frogs that reproduce in Balaour pond leave the water after spawning to reach their surrounding terrestrial summer habitats. We did not catch adult frogs on land, so we do not know the infection status of these adults.

One recently hatched tadpole, i.e., hatched for less than one week in the prevailing environmental conditions at this altitude (water temperature about 6 °C, Figure 2 and [29]), of the five tested was Rv+ (Table 2). Thereafter, live tadpoles were Rv+ until metamorphosis (Session 6; 19 September). The prevalence of *Ranavirus* increased with time, reaching 100% (*n* = 5/5) at Session 5 (15–16 August), and then dropped to 20% (*n* = 1/5) at Session 6 (19 September) (Figure 2). The variation in mean GE across the tadpole developmental period shows a rather low viral load until Session 4 (27 July) in live tadpoles, while the load in dead tadpoles was higher by an order of magnitude of 7 (10^7^). After this date, the GE values in live tadpoles stayed high (10^4^).

None of the terrestrial and aquatic insects collected from June to September tested positive for *Ranavirus*.

### 3.3. Ranavirus Detection in Water

Water samples from the first two sampling sessions were Rv- (Figure 2). From Session 3 (8 July) to Session 5 (15–16 August), the 12 qPCR replicates performed for each water sample were Rv+. *Ranavirus* was still detected at Session 6 (19 September), though it had the lowest detectability (5/12 replicates). The number of DNA copies (Figure 2) reached its maximum on 27 July (by an order of magnitude of 2), when the dead tadpoles were also observed with a high GE load. However, the GE values in infected live tadpoles and the number of DNA *Ranavirus* copies in water were not significantly correlated (Spearman rank correlation *R* = 0.25, *p* > 0.05).

### 3.4. Ranavirus in Sediment

All the qPCR replicates were negative for the samples collected at Sessions 1, 2, 4, and 6. At Sessions 3 and 5, 1 qPCR of the three was positive for both samples; these two samples were rerun. Only one positive replicate was observed again, so the two sediment samples were also considered negative.

### 3.5. Ranavirus Identification

The sequences obtained from the DNA extracted from dead tadpoles (*n* = 5), live tadpoles (*n* = 5), dead adults (*n* = 1), and live adults (*n* = 3) were 100% identical to the CMTV (GenBank accession number JQ231222) isolated from a common midwife toad (*Alytes obstetricans*; [12]) and an alpine newt (*Ichthyosaura alpestris*) in Spain [36]. This *Ranavirus* has been identified as the etiologic agent of the mass mortality of the common frogs observed in the region [13].

## 4. Discussion

### 4.1. Seasonal Dynamics of Ranavirus Epidemics in a Common Frog Population

The common frogs (tadpoles and adults, dead and alive) testing positive for *Ranavirus* (CMTV, [12]) in Balaour pond in 2016 confirm the widespread distribution of this pathogen in the south-eastern Alps [13].

At the first sampling session (10 June), egg-masses had been deposited in the pond, but only 16 adults were observed in the water, and they did not exhibit breeding behavior. Spawning in the common frog is synchronous and short in duration [37]. On this date, one dead adult was observed in the water. This specimen was Rv+. Of the live adults, 5 specimens were Rv-, while 1 out of 5 live tadpoles was Rv+. Of the few adults that remained in the pond until July, some were Rv+, but no further adult mortality was observed in the pond. In contrast, tadpoles suffered mass mortality at the end of July, when they reached developmental stages 35–39. *Ranavirus* was not detected in the sampled insects (aquatic or terrestrial), nor in the sediment collected during the summer activity period.

Sampling different life stages of organisms, as well as biotic and abiotic components, is necessary to provide a comprehensive view of *Ranavirus* dynamics (e.g., [38]). The observed dynamics in Balaour pond raise several questions: *Ranavirus* (including CMTV) is often highly pathogenic, and adult mass mortalities in common frog populations have been described in this region [13]. Yet adult mass mortality was not observed in Balaour pond in 2016. The breeding population is estimated to consist of at least 400 breeding adults (based on the observation of approximately 200 egg-masses in the pond and assuming an unbiased sex ratio [39]). Frog mass mortality is easily detectable in these small alpine ponds, so it is unlikely that an adult die-off went undetected. While frog aggregation during breeding can potentially foster pathogen transmission [40,41], the small fraction of adults that remained in the water after breeding was not infected at the beginning of the activity season. We have no data on the infection status of adults coming to breed or leaving the water after breeding, and the source and timing of the *Ranavirus* introduction remain unknown. It is unlikely that *Ranavirus* persists in the pond water from one year to the next [42], so it may be that sub-lethally infected adults (hibernating in water or on land) expose hatchlings to *Ranavirus* each year. Other reservoir species may also contribute to seasonal epidemics [43]; *Ranavirus* has been detected in fish in neighboring lakes [13]. However, the studied pond is free of fish, and the other communities we tested (aquatic and terrestrial insects) were Rv-. The characteristics of *Ranavirus* persistence from year to year in this pond remain to be studied.

In contrast to adults, tadpoles did suffer mass mortality. Rapid and synchronous mass mortality of tadpoles in frog populations is well known [38]. The sudden introduction of a virus can be the cause of such a die-off, but this does not seem to be the case in Balaour pond, as *Ranavirus* was detected from the very beginning of the activity season. The existence of a window of host susceptibility mediated by environmental conditions (e.g., temperature) may contribute to this pattern (e.g., [38]). *Ranavirus* epidemics often occur during late spring or summer and can begin and end within weeks [38,44,45,46,47]. Infected tadpoles in Balaour pond were observed throughout the activity season (from June to September), i.e., from early developmental to pre-metamorphosis stages, while tadpoles died in mass only at the end of July, at stages 35–39 (Figure 2). Hall et al. (2018) [38] observed that *Ranavirus* prevalence reached high levels (>50%) in wood frog tadpoles up to six weeks before mass mortality. High prevalence was also observed in the common frog population one month before the mass mortality in our study. In the wood frog [38], mortality occurred when tadpoles reached developmental stages (hind limb formation) that coincide with higher water temperatures (>15 °C). Several studies indicate that the pathogenicity of *Ranavirus* depends on the individual’s developmental stage, with the most susceptible stage varying between species [10,11,43,44]. In our study, the tadpole die-off was found to coincide with the highest virus load in tadpole tissue (Figure 2, [48,49]). The accumulation of infectious dead tadpoles may also facilitate transmission (Figure 1, [50,51,52]). Water temperature, independent of developmental stage, has been shown to increase virus pathogenicity [53]. Common frog tadpoles suffer greater mortality at 20 °C than at 15 °C, whether exposed to *Ranavirus* (FV3) or not [54]. The mass mortality of tadpoles observed in Balaour pond corresponded with the highest temperature recorded during the sampling period (Figure 2, 20 °C at the surface). However, no mortality was observed at the beginning of July or the end of August, when similar high temperatures were recorded. The respective and interacting roles of water temperature and tadpole developmental stage on *Ranavirus* pathogenicity remain to be studied in this common frog population.

### 4.2. Designing Pathogen Surveys Using eDNA and Occupancy Models

Diagnosing *Ranavirus* infection in amphibians requires collecting samples from either live individuals (e.g., a piece clipped from the tail or toe or skin swabs [55,56,57]) or dead individuals (e.g., a sample from the liver) [55]). However, *Ranavirus* can also resist adverse conditions (e.g., the drying out or freezing of a host carcass) and can then be shed into the water from infected individuals [58,59]. While viruses are rapidly degraded by microbes and zooplankton predation in water [42], they can remain detectable for at least seven days [60]. Water sampling and eDNA testing can thus indicate the presence of the pathogen; Hall et al. (2016) demonstrated that in ponds where wood frog tadpoles were Rv+, collected water samples also revealed the presence of *Ranavirus* DNA [24]. Other pathogens have also been detected using water samples—several metazoan parasites [26,61,62], chytrid fungus [21,22,23,24,25,27,63,64], and the Oomycota fungus *Aphanomyces astaci*, the causative agent of crayfish plague [28]—demonstrating the usefulness of eDNA for detecting pathogen infections in wildlife.

Table 2 summarizes different methods used to collect water for pathogen detection (Chytrids and *Ranavirus*). The most usual method is to collect water samples that are then filtered in the field. The filters are then stored until DNA extraction and amplification in the laboratory. The sampling design varies, e.g., discrete samples may be filtered separately or combined before filtering (Table 2). The location of the water sampling also differs, from rather exhaustive sampling (e.g., at approximately equidistant locations around the site) to solely where the presence of the pathogen would be expected (e.g., locations where frogs or tadpoles have been present). Water collection and filtering procedures are simple and have been used by non-experts in a citizen science program [65]. Clearly, the detection of a pathogen in water samples depends on its density (DNA quantities) and distribution (e.g., continuous or patchy) in the studied site. Several estimates concerning sampling effort are available; Julian et al. (2019) evaluated that as few as five water samples (Table 2) taken in June or July can detect both Bd and *Ranavirus* with 95% confidence [65]. In another study, every pond with *Ranavirus*-infected tadpoles tested positive with just three 250 mL water samples (Table 2, [27]). In our study, *Ranavirus* detection from water samples was effective in four out of six sessions, i.e., 66.6% detectability. In contrast, mass mortality was observed on only one date, i.e., 16.6% detectability. When live tadpoles were collected and molecular diagnostics performed, the presence of the pathogen was detected throughout the six sampling dates (100% detectability). Increasing the number of water samples will increase the cost of the molecular diagnostics, but pooling the samples before filtration allows a good sampling coverage of the study area at a reasonable cost. When the distribution of the pathogen (or the potential hosts) is known, we recommend to perform a pilot study to optimize the number (or volume) of water samples. Without preliminary knowledge, the screening of the site can be based on samples spaced evenly along the entire site circumference, then combined in one water samples which is then filtered to collect DNA.

Nonetheless, most pathogen detection tests are imperfect, resulting in false negatives when a pathogen is present but not detected. False negatives can occur when specimens or water samples are collected in the field, from organs in infected hosts (histological diagnostics) or in PCR replicates (molecular diagnostics). To address this, multi-scale occupancy models [66] can be used to develop a species distribution model to estimate the proportion of sites where a species or a pathogen occurs [67], even when detection is imperfect. This statistical framework is particularly suitable for eDNA analyses in which replicates (and thus detection probability) are present at the different steps of the method [68]; for example, it has been recommended in disease ecology [69]. Table 2 provides some studies that used eDNA to detect Bd or *Ranavirus* as well as occupancy models to estimate pathogen prevalence. Naïve prevalence (i.e., estimated prevalence without taking into account imperfect detection) can lead to the underestimation of Bd prevalence in ponds [23]; estimating correct prevalence requires taking multiple samples per site (Table 2). Replicates to evaluate detection probabilities can be temporal (i.e., sampling the same site at least three occasions) or spatial (i.e., sampling in at least three locations in the site) [66].

## 5. Conclusions

Inferring the health status of common frogs or ecologically similar amphibian species based only on the observation of mass mortality events in breeding sites runs the risk of missing potential infections. A better strategy is to sample amphibian tissue (adults and/or tadpoles) during, for example, the tadpole developmental phase and then perform molecular diagnostics. As this requires invasive techniques, administrative authorizations for protected species and/or protected areas and specific materials to collect and store the samples are needed. In some cases, catching frogs and tadpoles can be challenging, and tissue sampling requires a certain level of technical competence (and, sometimes, certified training). In light of these challenges, our results confirm previous findings that eDNA can be an effective alternative, since the temporal window for detecting the pathogen is rather large, collecting water samples is simple and can be performed by most field technicians, and, moreover, no specimen has to be caught, manipulated, or sacrificed. Moreover, applying a sampling design based on occupancy modelling (e.g., replicates allowing the evaluation of detection probability) provides prevalence estimates that are comparable between sites or years. As a biodiversity inventory method in marine and freshwater environments, eDNA has been extraordinarily successful; likewise, it holds clear potential for improving disease surveillance programs and increasing the performance of wildlife health status monitoring.

## Figures and Tables

**Figure 1 viruses-11-00526-f001:**
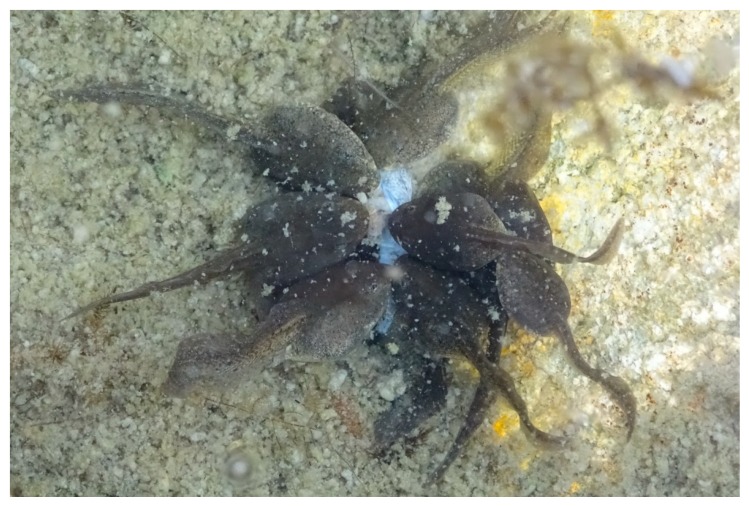
Common frog *Rana temporaria* tadpoles feeding on their dead congeners (photo L. Miaud).

**Figure 2 viruses-11-00526-f002:**
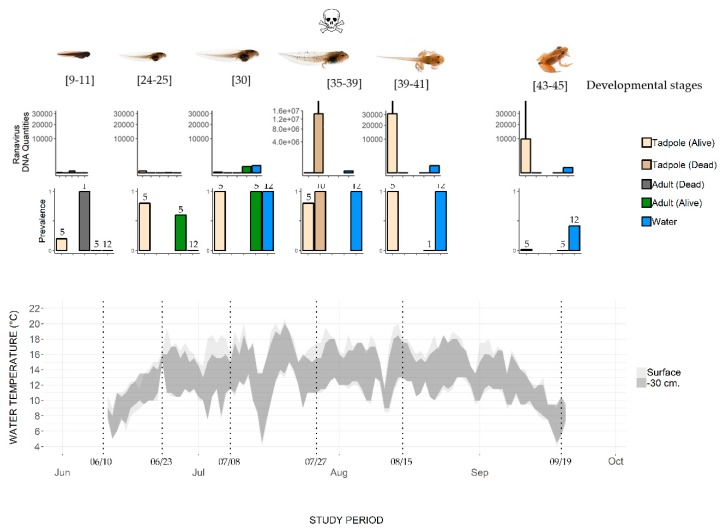
Change in water temperature and *Ranavirus* prevalence and load in a pond where the mass mortality of common frog tadpoles was observed. Water temperature was recorded at the surface and 30 cm below the surface. Top row of histograms: *Ranavirus* DNA quantities in common frogs (adults and tadpoles) and water samples (log scales, mean value, and SD). Bottom row of histograms: Prevalence (number of *Ranavirus* positive specimens/total number of sampled specimens, with sample size indicated above the bars). Developmental stages are based on Gosner stages; mortality was observed only on July 27.

**Table 1 viruses-11-00526-t001:** Sampling at Balaour pond in 2016.

Samples	Date					
	10–11 June	23–24 June	08 July	27 July	15-16 August	19 September
Tadpole alive	5 (9–11) ^1^	5 (24–25)	5 (30)	5 (35–39)	5 (39–41)	5 (43–45)
Tadpole dead	-	-	-	10 (35–39)	-	
Adult alive ^2^	5	5	5	0	1	5
Adult dead	1	-	-	-	-	-
Terrestrial insects ^3^	1	1	1	1	1	1
Aquatic insects ^4^	1	1	1	1	1	1
Sediment ^5^	1	1	1	1	1	1
Water ^6^	1	1	1	1	1	1

^1^ Stage of development (based on Gosner stages [30]); ^2^ the distal phalange of the second forelimb toe was collected from adults caught in water; ^3^ one sample per date, consisting of all flying insects caught with three yellow plates placed around the pond and then pooled together; ^4^ one sample per date, consisting of all aquatic insects caught by dip-netting in the pond and then pooled together; ^5^ one sample of sediment per date, consisting of 20 subsamples collected around the pond and then pooled together; ^6^ one sample of water per date, consisting of 20 subsamples collected around the pond, filtered and pooled together (see M&M for more details).

**Table 2 viruses-11-00526-t002:** Water sampling for Chytris and *Ranavirus* detection.

Pathogen	Filter Mesh Water Collection	Volume of Water Sample	Field Sampling	Date	Occupancy Design	Reference
Chytrid	0.45 µm Peristaltic pump	0.05 to 2.3 L per site (filter clogged)	In shallow water (0.1 to 0.75 m deep) in known or likely amphibian habitats. 3 locations per site, 4 sites	1 date	no	[21]
Chytrid	0.45 µm 50 mL syringe	<1 L per site (filter clogged)	Within 10 cm of the edge 1 location per site, 42 sites	1 date	no	[22]
Chytrid	0.22 µm 60 mL syringe	600 mL per site	Samples spaced evenly along the entire site circumference, but taken only from areas where frogs or tadpoles were present 30 samples of 20 mL combined, 20 sites	4 dates	No, yes *	[23,63]
Chytrid	1.2 µm Hand pump	500 to 1500 mL per site	Every 40 m along the shoreline. 5 samples of 50 mL combined, 13 sites	1 date	no	[26]
Chytrid	0.22 µm 60 mL syringe	20 mL to 2.4 L per site	In shallow water (5 and 20 cm below the water surface) 3 spatial replicates per site, 41 sites	1 date	yes	[64]
*Ranavirus*	0.2 µm Disposable paper cup	750 mL 250 mL × 3 per site	At 3 distinct locations (north, east and west) along the shore and surface (ca. 10 cm deep) 3 spatial replicates per site, 20 sites	1 date	yes	[24]
*Ranavirus*	0.2 µm Disposable paper cup	750 mL 250 mL × 3 per site	As above 8 sites	16 dates	no	[27]
Chytrids *Ranavirus*	0.2 µm Hand pump	0.6–1 L 150−250 mL × 4 per site	4 locations approximately equidistant around the site (within 2 m of the edge and 20−40 cm below the surface), 4 spatial replicates per site; 21 sites	3 dates	yes	[65]
*Ranavirus*	0.45 µm Sterile water-sample dipper	2 L 100 mL × 20 combined	20 locations equidistant around the site (within 0.5 m of the edge and 10 cm below the surface), 1 site	6 dates	no	This study

* the same data set was used in the two studies.

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
