# Peer review of "eDNA Increases the Detectability of Ranavirus Infection in an Alpine Amphibian Population"

_viruses, 2019, doi:10.3390/v11060526_

Round 1

Reviewer 1 Report

Miaud et al: Using eDNA to monitor Ranavirus in amphibian populations

In this manuscript the authors present data on ranavirus infection in several types of samples across six sampling periods in one pond. They test tissues from tadpoles and adults as well as from insects, and collect eDNA from water and sediment. Moreover, their sampling encompassed a tadpole die-off. Bringing together multiple types of information is very useful. Combined, these results provide a window onto the dynamics of ranavirus infection within a pond, which is useful for those interested in ranavirus epidemiology. However, this is not the way the paper is presented. Indeed, it was not clear to me what the author's question or point were. Based on the title and introduction they seem most interested evaluating eDNA-based sampling, but they do not do this well or transparently, and  it ha already done, elsewhere. Their results and discussion focus more on the ranavirus dynamics, but it is unclear to me what questions they were trying to answer or what one can conclude. I suspect this is a case of the authors being too close to their own system and data. They need to gain some perspective and decide what this paper is about and what an outside reader needs to know, and then revise the paper accordingly. Personally, my advice would be to treat this  as a case study of ranavirus dynamics in a place where a die-off occurred, but with much more care about what their data can and cannot say. 

General comments

The paper suffers from a lack of organization and clarify in presentation. The introduction is overly long and rambling. It seems to introduce the topic from the point of 1) amphibian decline, 2) the need for rapid identification, 3) the need to sample more than the target species, and 4) molecular tools more broadly. None of these seem to relate to the first aim of your study (L72-73) and only tangentially relate to the second aim. In the end, I want, but do not have a good sense of theme of the paper or study. I recommend a more direct, concise, focused introduction that leads to one or two specific research questions.  Both the methods and results are poorly organized and overly long. In both cases I had to search in multiple places to answer simple questions about who was sampled or whether different types of samples were similarly positive at a given time. Figure 2 was a good start at making these basic comparisons possible, but I would suggest adding to it and removing other less useful presentations (e.g., the tables). The discussion is a jumble of ideas, even within a paragraph. At the end I do not know what to conclude in general. The authors seem to have missed the chance to make any strong, defensible conclusions about any aspect of ranavirus biology or eDNA sample. 

Specific comments

L14-15: This clause, "...so the infection status..." does not make sense. A population is a collection of members of a given species. Please clarify or remove.

L15-16: This sentence is awkward and unnecessary. Consider removing.

L22: I was curious about the phrase, "only one adult was found dead." Were you expecting more mortality? If so, why?

L23: Change "virus" to "viral DNA"

L28: I am confused by the phrase, "in which the temporal window for detecting the pathogen is rather large." Is this meant to imply that eDNA is not effective with a narrow window of detection? What is the window of detection?

 L30-31: I do not disagree with this statement, but it is not supported by the results presented in the abstract. There are no statements about performance or accuracy. 

L35-41: Most of this seems unnecessary to introduce your study. Consider removing or condensing.

L51-52: I do not disagree with this sentence, but it is a bit disingenuous. There are, as far as I know, no "appropriate mitigation measures" for ranavirus outbreaks and the reference is for a paper about human public health infrastructure. 

L54-57: The logic here is not entirely clear. First, it is not clear how sampling other species provides information about infection in a focal species that is not provided by simply sampling the focal species. Second, asymptomatic carriers could certainly occur in the "impacted host species." To me, the value of eDNA seems to be that it can sample from virus shed across a community of hosts, both heavily infected and, at least theoretically, asymptomatic carriers, although this has not be established so far as I know. 

L65-66: What do you mean by "contamination"? Moreover, the second sentence of this paragraph has no clear connection to this first sentence.

L73: "carrier" has a specific meaning. I think you simply mean "host". 

L79-83: Why are you describing lakes that we're not sampled in this study? 

L87: Use commas around "Rana temporaria"

L104-118: There is a great deal of replication within this paragraph as well as with Table 1. Please write the general methods here (e.g., how the sampling was conducted) and then include the results in the table and, as necessary, the results section. It was not clear that samples were stored individually or in batches, which could lead to contamination. It was also not clear water steps the researchers took to avoid cross contamination (e.g., using gloves, disinfecting equipment between animals, etc.). Please provide these details.  

Replace "°" with "%" when referring to a percent. Also, am I correct to think this alcohol is ethanol?

L134: I appreciate knowing this detail, but there is no reason to think that 95% ethanol would inactivate DNA, so external DNA contamination could be moved from one insect to the next (although with no positive samples, this is not an issue).

L140: Were water samples collected prior to animal samples? That is, standing on the pond bank to collect water does no good if researchers were already in the pond to collect animals. I assume the researchers did this correctly, but the order of presentation leaves me with the question.

L142-144: Was the entire 2 L of water pushed through the filter, or just 100 mL? Please clarify. Also, I was curious about the VigiDNA filter, but I could find no details. Spygen seems to offer services, but does not appear to provide consumable products like filters. Please clarify the details of the filter then so others can evaluate it or try to follow your approach. 

L159-162: I am confused by how much tissue went into the extractions. Even a small tadpole would be enough to overwhelm most extraction kits. The "bulk of insects" would be far, far too much material! I see no indication that there was a check to ensure DNA was obtained with these methods, which leaves open the possibility that negatives were due to poor DNA extraction, at least from insects. These details can have a strong effect on DNA quality and quantity and thus the ability to detect ranavirus. 

L164: Please clarify that "three replicates" means each sample was run in triplicate qPCR reactions. Also, what was the positive control?

L177-182: Why were was the DNA from sediment sequenced? Why not other types of samples? Very confusing.

L187: My understanding is that a stronger bleach solution (e.g., 50% ) is necessary to effectively degrade DNA.  

L200: amplification of what? A synthetic exogenous standard or something else? 

L201-213: It is unclear whether these methods apply to all samples or just to the water samples. I appreciate this level of detail, but I would like to know them for all types of samples. 

L215: Does the mean number of DNA copies include the zeros from negative wells?

L218-220: This is useful, but should perhaps be part of the site description above.

L221-239: This is useful, but repetitive of the methods. Please consolidate the results in one place. 

Figure 2: I think that figure 2 is a good start at presenting the results in a clear, concise way. I would suggest building on this figure to provide details found in tables 1 and 2. For instance, it would be simple to provide larval and adult sample sizes in the figure as well as Gosner stages. Figure 3 could be incorporated as a panel in Figure 2 showing the intensity of infections (GE) in the samples.  My preference is for points on a graph connected by lines rather than side-by-side bar plots, but this is a minor point. In any case, I think showing data in a single figure makes comparison much simpler and I think table 2 should be moved to supplemental data.

L254: What compartments do the authors mean? I would include insects as a compartment, but virus was not detected in them, yes? So this is confusing.

L263-265: Is the point that the frogs in the pond were abnormal? Or something else? This is a non sequitur. Also, do you mean "infection status" instead of "contamination status"? 

L270-273: The information in viral loads would be much easier to understand in a graph.

L287-8: This is a bit misleading. The "replicates" were replicate wells of a qPCR reaction, but they all came from the same DNA extraction of the same 2L sample. In this case, I am struggling to understand what information is provided by the proportion of the 12 wells with amplification that is not directly related to the quantity of viral eDNA in the sample. 

L292-293: Why are the results of ranavirus detection in sediments included in the figures and table?

L295: I wonder if the part about sequencing from sediment samples was misplaced and should have gone with the part about detecting virus in animals. 

L303: replace "(" with a comma.

L305: fix citations

L307: This assertion is not supported by at least one of the references. It is true in theory, but I do not know that this is supported empirically. 

Also, this paragraph jumped between ideas without transition or context. It is very confusing.

L316: In contrast to what? The previous idea was about persistence. 

L318: many of these papers referenced do not support this statement or have anything to do with it.  I would suggest looking at two papers for additional context and possible citation:

Gahl, M. K., and A. J. K. Calhoun. 2010. The role of multiple stressors in ranavirus-caused amphibian mortalities in Acadia National Park wetlands. Canadian Journal of Zoology 88:108-121. 

Hall, E. M., C. S. Goldberg, J. L. Brunner, and E. J. Crespi. 2018. Seasonal dynamics and potential drivers of ranavirus epidemics in wood frog populations. Oecologia 188:1253-1262. 

L320-323: To be clear, I do not think anyone is suggesting metamorphosis is the most susceptible stage, but pro-metamorphic stages in at least wood frogs. In any case, I do not think your data are sufficient to discern which stages are most likely to die from ranavirus infections. You have small sample sizes, somewhat infrequent sampling, and do not track individual fates. Your data are not incompatible with the hypothesis that pro-metamorphic stages are most likely to die. 

L327-329: Figure 2 suggests the highest temperatures were noted a week or more before you observed dead tadpoles. It is certainly possible the die-off was trigged by high temperatures, but it is a too much to say that the die-off coincided with this high temperature. 

Also, I think you can address the temperature and stage hypotheses at the same time more efficiently. All you know for certain is that the die-off occurred some time after the high surface temperature and when animals were stages 35-39, but you cannot be certain of much more than that. 

L330: Your sampling doe not let you say anything about the rapidity of the die-off. It could have been going on for at least a week or just before you arrived to sample. 

L333: I am surprised you did not mention the dead infected adult found when eggs were present. Was it is possible this animal was infected before returning to the pond to breed or that it died the previous year and was frozen? In any case, it seems a likely source of virus to the adults and tadpoles. 

L338-339: This is not true. One need not find a die-off to detect ranavirus. 

L340-41: Care must be exercised when analyzing any sample, including tissues and eDNA. Indeed, one is even less certain what eDNA means relative to infection in animals! 

More broadly, I am not certain at all what the paragraph is arguing. 

L350: If you are going to make comparisons between viral titers in the animals and eDNA samples, do so formally (e.g., a simple correlation). The current approach uses many words to say very little. 

L373: What does "infection range" mean? Following that, is anyone arguing that only trying to detect die-offs is a good means of surveillance? I do not think so. You are arguing against a straw man.

Author Response

Response to Reviewer 1 Comments

Point 1: In this manuscript the authors present data on ranavirus infection in several types of samples across six sampling periods in one pond. They test tissues from tadpoles and adults as well as from insects, and collect eDNA from water and sediment. Moreover, their sampling encompassed a tadpole die-off. Bringing together multiple types of information is very useful. Combined, these results provide a window onto the dynamics of ranavirus infection within a pond, which is useful for those interested in ranavirus epidemiology. However, this is not the way the paper is presented. Indeed, it was not clear to me what the author's question or point were. Based on the title and introduction they seem most interested evaluating eDNA-based sampling, but they do not do this well or transparently, and it had already done, elsewhere. Their results and discussion focus more on the ranavirus dynamics, but it is unclear to me what questions they were trying to answer or what one can conclude. I suspect this is a case of the authors being too close to their own system and data. They need to gain some perspective and decide what this paper is about and what an outside reader needs to know, and then revise the paper accordingly. Personally, my advice would be to treat this  as a case study of ranavirus dynamics in a place where a die-off occurred, but with much more care about what their data can and cannot say. 

Response 1: We partly agree to the reviewer’s comment.

Yes, the paper describe a longitudinal study of ranavirus infection in one place, taking into account the different ecosystem compartments. Yes, the paper focus on the detection method, especially by comparing tissue versus water (eDNA) sampling.

Reviewer ‘comment “they seem most interested evaluating eDNA-based sampling, but they do not do this well or transparently, and it had already done, elsewhere”. Yes, eDNA has been used to detect ranavirus (and several other pathogens), but in the only paper we know, the design is very different: the authors used eDNA in ponds where and when infection are present (with die-off), and collected tissue from tadpole and dead adult of one amphibian species. In our manuscript, we collected samples (different compartments, dead and live specimen) along the activity season, in order to compare ranavirus detection even when mortality does not occur (i.e. before and after the die-off).

We consider the interest and originality of this manuscript is exactly in combining the longitudinal sampling and different detection methods. As reviewer 2 asked also ask to make “more concrete recommendation” in the discussion chapter, we will keep the “methodological” angle of view of this manuscript.

Point2: The paper suffers from a lack of organization and clarify in presentation. The introduction is overly long and rambling. It seems to introduce the topic from the point of 1) amphibian decline, 2) the need for rapid identification, 3) the need to sample more than the target species, and 4) molecular tools more broadly. None of these seem to relate to the first aim of your study (L72-73) and only tangentially relate to the second aim. In the end, I want, but do not have a good sense of theme of the paper or study. I recommend a more direct, concise, focused introduction that leads to one or two specific research questions.

Response 2: We partly agree with this comment and make the following changes:

We shorten the introduction (suppressing references to Chytrids), focusing only directly to ranavirus (we however keep the new reference proposed by the reviewer 2, Scheele et al., 2019, Science).

We keep the paragraph on Rapid and accurate pathogen identification including molecular tools. We introduce the use of eDNA by describing the only available study using this method to detect ranavirus.

Finally, we keep the description of the first aim (1) to describe the infection status……but we change the formulation of the second aim by (2) compare sampling methods (from organisms to water) to improve ranavirus detectability, and provide some guidelines for a better implementation of virus surveillance programmes.

 Both the methods and results are poorly organized and overly long. In both cases I had to search in multiple places to answer simple questions about who was sampled or whether different types of samples were similarly positive at a given time.

See Specific comments

Figure 2 was a good start at making these basic comparisons possible, but I would suggest adding to it and removing other less useful presentations (e.g., the tables).

We agree and follow the specific comments on this fig and tables.

The discussion is a jumble of ideas, even within a paragraph. At the end I do not know what to conclude in general. The authors seem to have missed the chance to make any strong, defensible conclusions about any aspect of ranavirus biology or eDNA sample. 

We found this last sentence rather severe…because we produce results on ranavirus biology (which hosts or substrates) and eDNA sampling. Whatever, we try to highlight better these results in the discussion (see specific comments).

Point 3: L 14-15: This clause, "...so the infection status..." does not make sense. A population is a collection of members of a given species. Please clarify or remove.

Response 3: We change the sentence to” Viruses in the Iridoviridae family, such as those in the Ranavirus genus, can infect amphibian species without resulting in mortality or clinical signs, and they can infect other hosts than amphibian species.

Point 4: L15-16: This sentence is awkward and unnecessary. Consider removing.

Response 4: Done

Point 5: L22: I was curious about the phrase, "only one adult was found dead." Were you expecting more mortality? If so, why?

Response 5: No, we were not expecting more. We remove “only” from the sentence.

Point 6: L23: Change "virus" to "viral DNA"

Response 6: Done

Point 7: L28: I am confused by the phrase, "in which the temporal window for detecting the pathogen is rather large." Is this meant to imply that eDNA is not effective with a narrow window of detection? What is the window of detection?

Response 7: We agree to supress this expression (temporal window of detection) in the summary. We change the sentence to: “Our results indicate that eDNA can be an effective alternative to tissue/specimen sampling, and can detect ranavirus presence outside die-offs.”

Point 8: L30-31: I do not disagree with this statement, but it is not supported by the results presented in the abstract. There are no statements about performance or accuracy. 

Response 8: We added “eDNA…… can detect ranavirus presence outside die-offs.” As eDNA can detect ranavirus before, during and after die-off, and does not require tissue sampling, we consider that the concluding sentence of the summary can be kept.

Point 9: L35-41: Most of this seems unnecessary to introduce your study. Consider removing or condensing.

Response 9: We agree and as said in the General comments (response 2), we shorten the introduction by removing these sentences.

Point 10: L51-52: I do not disagree with this sentence, but it is a bit disingenuous. There are, as far as I know, no "appropriate mitigation measures" for ranavirus outbreaks and the reference is for a paper about human public health infrastructure. 

Response 10: We agree with this comment. The reference is for human health but illustrate the way to implement mitigation measure. This could be confusing (it is not mitigation measure for ranavirus. As it is a very general statement, we propose to keep it as it well introduce the following sentences, but simply supress the reference.

Point 11: L54-57: The logic here is not entirely clear. First, it is not clear how sampling other species provides information about infection in a focal species that is not provided by simply sampling the focal species. Second, asymptomatic carriers could certainly occur in the "impacted host species." To me, the value of eDNA seems to be that it can sample from virus shed across a community of hosts, both heavily infected and, at least theoretically, asymptomatic carriers, although this has not be established so far as I know.

Response 11: We clarify the paragraph by changing the sentences as:

Several field studies have reported ranavirus-infected amphibians that do not present clinical signs or histological changes (e.g. [18]). Transmission of ranarirus is possible between amphibians, reptiles and bony fish, potentially playing the role of a pathogen reservoir [19]. Evaluating the individual risk of infection in an amphibian population thus requires sampling all these potentially infected hosts.

Point 12: L65-66: What do you mean by "contamination"? Moreover, the second sentence of this paragraph has no clear connection to this first sentence.

Response 12: We agree and suppress the sentence “Environmental sampling to detect ranavirus may potentially identify the different sources of contamination”, and move the second sentence in the previous paragraph.

Point 13: L73: "carrier" has a specific meaning. I think you simply mean "host".

Response 13: We agree and change “carrier” to “hosts”

Point 14: L79-83: Why are you describing lakes that we're not sampled in this study?

Response 14: We agree and suppress the lakes’ description

Point 15: L87: Use commas around "Rana temporaria"

Response 15: Done

Point 16: L104-118: There is a great deal of replication within this paragraph as well as with Table 1. Please write the general methods here (e.g., how the sampling was conducted) and then include the results in the table and, as necessary, the results section. It was not clear that samples were stored individually or in batches, which could lead to contamination. It was also not clear water steps the researchers took to avoid cross contamination (e.g., using gloves, disinfecting equipment between animals, etc.). Please provide these details.

Response 16: We agree and change the table 1 by adding tadpole stage of development and the results of the sampling (also according to reviewer’ 2). We change the paragraph as follow:

“The pond was visually scan to detect tadpoles and adults resting on the bottom. Live tadpoles were caught by dip-net (mesh 1mm). They were stored in individual plastic bottle (1.5 ml) filled in with 95% ethanol. Dead tadpoles were collected and stored similarly. Live adults were caught by dipnet (5mm mesh). The distal phalange of the second forelimb toe was collected and stored individually in plastic bottle (1.5 ml) filled in with 95% ethanol. Disposable gloves were used and changed between each sampling, and the material (scissors and pliers) was soak in ethanol and go to the flame between each sample collection. The dead adult was collected with the dipnet and stored in a plastic bottle (300 ml) filled in with 95% ethanol. Table 1 provided the sample size and developmental stages [33] of tadpoles and adults collected. The toe clip allowed checking that unique adult specimens were sampled along the studied period”.

Point 17: Replace "°" with "%" when referring to a percent. Also, am I correct to think this alcohol is ethanol?

Response 17: We agree and change ° to % and alcohol to ethanol.

Point 18: L134: I appreciate knowing this detail, but there is no reason to think that 95% ethanol would inactivate DNA, so external DNA contamination could be moved from one insect to the next (although with no positive samples, this is not an issue).

Response 18: We agree and change this sentence as follows: “The aquatic insects collected at the same date were stored in the same 15 ml plastic bottle filled in with 95% ethanol”

Point 19: L140: Were water samples collected prior to animal samples? That is, standing on the pond bank to collect water does no good if researchers were already in the pond to collect animals. I assume the researchers did this correctly, but the order of presentation leaves me with the question.

Response 19: We fully agree! Of course, the water collection was the first sampling! Do we have to move the paragraph 2.5 to 2.1? If so, it will make a large change the line numbering, ie leading difficult to follow the changes. Perhaps this change can be done in a next draft?   

Point 20: L142-144: Was the entire 2 L of water pushed through the filter, or just 100 mL? Please clarify. Also, I was curious about the VigiDNA filter, but I could find no details. Spygen seems to offer services, but does not appear to provide consumable products like filters. Please clarify the details of the filter then so others can evaluate it or try to follow your approach.

Response 20: Yes, the whole volume (2 L) was filtered using the syringe several times. We change the sentence as: “The 2-L water sample was homogenized by gently shaking the bag to ensure the eDNA was evenly mixed throughout the sample, and the whole 2-L water sample was then filtered directly in the field through a VigiDNA 0.45 μm filter (SPYGEN, Le Bourget du Lac, France) using a sterile 100-mL syringe."

(Filters and sampling kits are sold under request at SPYGEN)

Point 21: L159-162: I am confused by how much tissue went into the extractions. Even a small tadpole would be enough to overwhelm most extraction kits. The "bulk of insects" would be far, far too much material! I see no indication that there was a check to ensure DNA was obtained with these methods, which leaves open the possibility that negatives were due to poor DNA extraction, at least from insects. These details can have a strong effect on DNA quality and quantity and thus the ability to detect ranavirus.

Response 21: We agree and we gave more details on this analysis. “For both aquatic and terrestrial insects, specimens were grouped per date of sampling and crushed together in 5 ml of 95° alcohol. A subsample of 1.5 mL was collected for DNA analysis”. For DNA checking, there is no issue with DNA extraction form insect as we are looking for ranavirus DNA. We are very confident with our result about ranavirus detection, as we always perform qPRC with ranavirus DNA standard obtained with plasmid DNA (dosed 6 ng/µl) and a dilution range from 6.622 E-17 g/µl to 6.622 E-10 g/µl).

Point 22: L164: Please clarify that "three replicates" means each sample was run in triplicate qPCR reactions. Also, what was the positive control?

Response 22: We clarify the text as follow: “All assays were performed in triplicate. Ranavirus DNA, provided by Stephen Price (Zoological Society, London, UK), was used as a positive control”.

Point 23: L177-182: Why were was the DNA from sediment sequenced? Why not other types of samples? Very confusing.

Response 23: We agree that this paragraph (2.9) has to be moved at the end of the results. We sequenced RNA collected from tadpoles, live and dead, adult (live and dead) and water. This is added in the new paragraph 2.10 Ranavirus identification:

Viral DNA was obtained from 1 dead adult, 3 live adults, 5 dead tadpoles and 5 live tadpoles. These 15 samples (530 base-pair PCR fragments) were sequenced using the BigDye Terminator Cycle Sequencing kit   …”

Point 24: L187: My understanding is that a stronger bleach solution (e.g., 50% ) is necessary

Response 24: This depends on the commercial bleach. In each country the household bleach have different Sodium hypochlorite concentration. To avoid any confusion we changed the following sentence to “All workbenches were decontaminated with commercial bleach, diluted to achieve 0.5% Sodium hypochlorite solution, before and after each manipulation”.

Point 25: L200: amplification of what? A synthetic exogenous standard or something else?

Response 25: We changed the sentence to “After DNA extraction, the samples were tested for inhibition using real-time amplification following the protocol described in Biggs et al., 2015. All the samples were found not inhibited." (same comments from Reviewer 2 and 3).

Point 26: L201-213: It is unclear whether these methods apply to all samples or just to the water samples. I appreciate this level of detail, but I would like to know them for all types of samples.

Response 26: It is for water samples, as it is in the 2.10 Ranavirus detection in water sample. We did this precise description as we developed this protocol. For the other samples (tissue, sediments, insects), we used published protocols.

Point 27: L215: Does the mean number of DNA copies include the zeros from negative wells?

Response 27: For eDNA, the 12 replicates were positive for 5 dates and the mean was calculated for these 12 replicates. On the 19 September, only 5 among the 12 replicates were Rv+ and the mean number of DNA copy was calculated without including the 7 zero values.

We change the sentence to “…… and mean number of DNA copies (using the number of copies per Rv+ replicates) for each sample.

Point 28: L218-220: This is useful, but should perhaps be part of the site description above.

Response 28: We propose to keep it in order to remain the ecological context.

Point 29: L221-239: This is useful, but repetitive of the methods. Please consolidate the results in one place.

Response 29: As we moved the method in table 1, this useful results stay at this place.

Point 30: Figure 2: I think that figure 2 is a good start at presenting the results in a clear, concise way. I would suggest building on this figure to provide details found in tables 1 and 2. For instance, it would be simple to provide larval and adult sample sizes in the figure as well as Gosner stages. Figure 3 could be incorporated as a panel in Figure 2 showing the intensity of infections (GE) in the samples.  My preference is for points on a graph connected by lines rather than side-by-side bar plots, but this is a minor point. In any case, I think showing data in a single figure makes comparison much simpler and I think table 2 should be moved to supplemental data.

Response 30: Following the reviewer’comment, we design a new figure with:

-       Gosner stages provoded in the figure

-       GE and number of DNA copies and SD) provided in the figure 2

-       Figure 3 incorporated in Figure 2

-       Table 2 should be moved to supplemental data.

Point 31: L254: What compartments do the authors mean? I would include insects as a compartment, but virus was not detected in them, yes? So this is confusing.

Response 31: We agree and change “all” by “several” (same comment as reviewer 2 point 10)

Point 32: L263-265: Is the point that the frogs in the pond were abnormal? Or something else? This is a non sequitur. Also, do you mean "infection status" instead of "contamination status"? 

Response 32: Leaving the pond after breeding is a normal behaviour, and some individuals staying in the water (or very close to it) is commonly observed. We change “contamination status” to “infection status”

Point 33: L270-273: The information in viral loads would be much easier to understand in a graph.

Response 33: The information (prevalence and load) are now available in the new figure 2.

Point 34: L287-8: This is a bit misleading. The "replicates" were replicate wells of a qPCR reaction, but they all came from the same DNA extraction of the same 2L sample. In this case, I am struggling to understand what information is provided by the proportion of the 12 wells with amplification that is not directly related to the quantity of viral eDNA in the sample.

Response 34: Yes, the replicates are replicate wells of the qPCR (explained in M&M paragraph 2.10). The information provided by this “prevalence” is link to the quantity of DNA of the focal species (ie ranavirtus) in the sample (also provided by the raw number of reads). Results are often presented in this way for data coming from eDNA inventories.

Point 35: L292-293: Why are the results of ranavirus detection in sediments included in the figures and table?

Response 35: We change the text (ranavirus detection in sediment is non-significant) so it is not presented in figure and table (same comment as reviewer 2, point 12).

Point 36: L295: I wonder if the part about sequencing from sediment samples was misplaced and should have gone with the part about detecting virus in animals.

Response 36: We did not sequence DNA from sediment

Point 37: L303: replace "(" with a comma.

Response 37: Done

Point 38: L305: fix citations

Response 38: Done

Point 39: L307: This assertion is not supported by at least one of the references. It is true in theory, but I do not know that this is supported empirically. Also, this paragraph jumped between ideas without transition or context. It is very confusing.

Response 39: We agree, the idea was as common frogs spawn in mass, there is a risk of higher transmission (cited in reference 41). We thus change the sentence to “

“Adult common frog aggregation during breeding can potentially foster pathogen transmission [41], but only on dead adult was observed in the Balaour pond. 

We suppress reference 42.

Point 40: L316: In contrast to what? The previous idea was about persistence. 

Response 40: We change the sentence to “In contrat to adults,….”

Point 41: L318: many of these papers referenced do not support this statement or have anything to do with it.  I would suggest looking at two papers for additional context and possible citation:

Gahl, M. K., and A. J. K. Calhoun. 2010. The role of multiple stressors in ranavirus-caused amphibian mortalities in Acadia National Park wetlands. Canadian Journal of Zoology 88:108-121. 

Hall, E. M., C. S. Goldberg, J. L. Brunner, and E. J. Crespi. 2018. Seasonal dynamics and potential drivers of ranavirus epidemics in wood frog populations. Oecologia 188:1253-1262. 

Response 41: We checked the reference to keep relevant paper. We change the sentence to

“Susceptibility to ranavirus depends on the individual’s developmental stage, and the most susceptible stage varies between species [12,13,15,45,46].

We suppress reference 15 and 45 from this list and added the two papers suggested by the reviewer

[15] Miaud, C.; Pozet, F.; Gaudin, N.C.G.; Martel, A.; Pasmans, F.; Labrut, S. Ranavirus Causes Mass Die-Offs of Alpine Amphibians in the Southwestern Alps, France. J. Wildl. Dis. 2016, 52, 242–252.

[45] Rijks, J.M.; Saucedo, B.; Spitzen-van der Sluijs, A.; Wilkie, G.S.; van Asten, A.J.A.M.; van den Broek, J.; Boonyarittichaikij, R.; Stege, M.; van der Sterren, F.; Martel, A.; et al. Investigation of Amphibian Mortality Events in Wildlife Reveals an On-Going Ranavirus Epidemic in the North of the Netherlands. Plos One 2016, 11, e0157473.

We would like to add this ref :

Hall, E. M., C. S. Goldberg, J. L. Brunner, and E. J. Crespi. 2018. Seasonal dynamics and potential drivers of ranavirus epidemics in wood frog populations. Oecologia 188:1253-1262. 

But we were technically not able to include it in the automatic adding (Zotero system). Could it be done for a next check with all the track change accepted (this will be ref 47, which wioll change all the following numbering)?

Point 42: L320-323: To be clear, I do not think anyone is suggesting metamorphosis is the most susceptible stage, but pro-metamorphic stages in at least wood frogs. In any case, I do not think your data are sufficient to discern which stages are most likely to die from ranavirus infections. You have small sample sizes, somewhat infrequent sampling, and do not track individual fates. Your data are not incompatible with the hypothesis that pro-metamorphic stages are most likely to die.

Response 42: We change the sentence to “In our study, Rv+ common frog tadpoles (without mortality or abnormal behaviour) were observed until metamorphosis. No tadpole mortality was observed before or after the July mass die-off, suggesting that the tadpole susceptibility could be stage-dependent [50]”.

Point 43: L327-329: Figure 2 suggests the highest temperatures were noted a week or more before you observed dead tadpoles. It is certainly possible the die-off was trigged by high temperatures, but it is a too much to say that the die-off coincided with this high temperature. Also, I think you can address the temperature and stage hypotheses at the same time more efficiently. All you know for certain is that the die-off occurred some time after the high surface temperature and when animals were stages 35-39, but you cannot be certain of much more than that.

Response 43: We agree and move the sentences about temperature to the paragraph on the relation with stage. The sentence was changed as: “The mass mortality of tadpoles observed in the Balaour pond follows the highest temperature recorded during the whole sampled period (Figure 2, 20 °C at the surface). However, no mortality was observed at the beginning of July or end of August where similar high temperatures were recorded. The respective and interactive role of water temperature and tadpole developmental stage on ranavirus infection remain to be studied in common frog tadpoles”.

Point 44: L330: Your sampling doe not let you say anything about the rapidity of the die-off. It could have been going on for at least a week or just before you arrived to sample. 

Response 44: Yes we agree, rapidity is a relative concept! We change the sentence (also answering here the point 45) as:

“Our longitudinal study showed that adults and tadpoles can be infected by ranavirus”. And “The dead frog (Rv+) found at the beginning of the breeding season could be one of these adults maintaining the infection in this population”

Point 45: L333: I am surprised you did not mention the dead infected adult found when eggs were present. Was it is possible this animal was infected before returning to the pond to breed or that it died the previous year and was frozen? In any case, it seems a likely source of virus to the adults and tadpoles. 

Response 45: We agree and add the information on this dead frog in the previous paragraph.

Point 46: L338-339: This is not true. One need not find a die-off to detect ranavirus.

Response 46: OK we change the sentence ((also taking into account point 13 of reviewer 2).to:

Several other pathogens have also been detected using water samples: several metazoan parasites [27,63,64], chytrid fungus [22–24,26,28,65,66,67,68], and Oomycota fungus

New ref:

68. Mosher, B. A., Brand, A. B., Wiewel, A. N., Miller, D. A., Gray, M. J., Miller, D. L., & Grant, E. H. C. (2018). Estimating occurrence, prevalence, and detection of amphibian pathogens: insights from occupancy models. Journal of Wildlife Diseases.

Point 47: L340-41: Care must be exercised when analyzing any sample, including tissues and eDNA. Indeed, one is even less certain what eDNA means relative to infection in animals! More broadly, I am not certain at all what the paragraph is arguing. 

Response 47: We agree and change the sentence (see point 46).

Point 48: L350: If you are going to make comparisons between viral titers in the animals and eDNA samples, do so formally (e.g., a simple correlation). The current approach uses many words to say very little. 

Response 48: We calculated the correlation and give the result in “result” section (3.3. Ranavirus detection in water)However, the GE values in infected tadpoles and the number of DNA ranavirus copies in water were not significantly correlated (Spearman rank correlation R = 0.25, p>0.05).

Point 49: L373: What does "infection range" mean? Following that, is anyone arguing that only trying to detect die-offs is a good means of surveillance? I do not think so. You are arguing against a straw man

Response 49: We want of course to argue for the opposite! We change the sentence to: “Inferring the health status of common frogs or ecologically similar amphibian species based only on the observation of mass mortality events in breeding sites runs a risk of missing potential infections.”.

Reviewer 2 Report

Summary:

The authors describe a study which aimed to investigate the presence of ranavirus in various “ecological compartments” before and during a die-off. They also say that their aim is to provide some guidelines to improve ranavirus survey methods and allow more effective virus surveillance programmes to be implemented. I found the paper to be well-written and interesting in terms of the first objective, but have noted several points that could be clarified to make the manuscript stronger. The second objective (providing ranavirus survey recommendations for surveillance programs) could be strengthened by spending more time in the discussion making concrete recommendations.

Title: Can you create a title that is more active and specific? Something that indicates what you found rather than the general topic would be valuable.

Introduction:

Lines 42-45: See new paper by Scheele et al. to update this information

Scheele, B. C., Pasmans, F., Skerratt, L. F., Berger, L., Martel, A., Beukema, W., ... & De la Riva, I. (2019). Amphibian fungal panzootic causes catastrophic and ongoing loss of biodiversity. Science, 363(6434), 1459-1463.

Methods:

Lines 79-86: I am confused about the description of the two lakes, as they were not sampled.

Lines 104-118 are almost completely redundant with Table 1. Can you include some additional information (e.g., Gosner stage) to the table and remove the text?

Lines 104-118: I assume that unique adult animals were sampled on different visits and that you could recognize this due to the toe clips, but it would be good to list this explicitly.

Line 143: This filter pore size is larger than what I have seen. How does it relate to the size of ranavirus particles? Or are you capturing animal skin cells that have ranavirus on them?

Lines 214-215: Based on your results it seems like prevalence was the number of positive replicates rather than the number of positive samples.

Results:

Did any of the negative controls come up positive? Was there any evidence of inhibition, especially in soil or water samples? (Your methods say that you tested for contamination and inhibition, but not what you found.)

Lines 254-255: This line is a little misleading because ranavirus was not detected in the insect community, which I would consider one of the “compartments of the studied ecosystem”.

Line 230: typo Most/mosu

Lines 286-290: Why were individual replicates considered positive/negative when using water sampling, but replicates were pooled to the sample level for the animal samples and sediment samples? (That is, if you had 1 of 3 water filter replicates that was positive you’d report a prevalence of 33% but if you had 1 of 3 animal replicates positive you’d call the sample negative. And you report 1/3 replicates being positive in the sediment analysis even though by your description in the methods [lines 167-168] it seems like the status of this sample would be ‘equivocal’.)

Discussion:

Lines 341-342: See recent manuscript by Mosher et al that investigates tail samples and finds they perform favorably.

Mosher, B. A., Brand, A. B., Wiewel, A. N., Miller, D. A., Gray, M. J., Miller, D. L., & Grant, E. H. C. (2018). Estimating occurrence, prevalence, and detection of amphibian pathogens: insights from occupancy models. Journal of Wildlife Diseases.

Line 357: Typo – asymptotic should be asymptomatic

I think that your manuscript needs to mention that ranavirus is generally detected imperfectly, wither due to sampling error (failing to capture DNA in certain samples – your pooling of water and sediment samples helps account for this), PCR inhibition, low quantity DNA, etc. Separating the detection probability from the occurrence of ranavirus requires modeling, but nested data like yours (multiple samples per site, multiple replicates per sample) are a natural fit for these occupancy models. I don’t believe that you need to re-run your analyses, but I do think it is prudent to recommend that readers consider the imperfect detectability in future studies. The Mosher et al. manuscript mentioned above introduces these ideas and, like you, concludes that there is great value in being able to sample for ranavirus using non-lethal methods outside the short window where die-offs occur.

Conclusions section: Can you make any concrete recommendations to others studying ranavirus in the field? How many water, sediment, or animal samples should folks be collecting to be certain of detecting ranavirus when present at various time points? I assume you need more animals or water samples early on, but how many? What laboratory methods do you recommend? Did you have contamination or PCR inhibition problems? What other suggestions might you have?

Figure 2: I suggest making the barplots larger in some way, and maintaining the same 4 categories (tadpole alive, tadpole dead, adult alive, and water sample) in each plot to help the reader. You can use an asterisk or something along those lines to indicate visits when certain categories were not detected/not sampled. Perhaps you can merge the barplot with the temperature data on a single plot with a second y axis? The inferences from the barplots seem to be more important than temperature differences to me, so those should be highlighted and easier to see.

Table 2: In my opinion this table should be appended.

Figure 3: The x-axis labels make it seem as though negative quantities were observed for some categories. Can you move the x-axis labels down and renumber the y-axis to start at 0, rather than at 0.1? Also, for estimates that are generated from multiple samples, please include a measure of precision (e.g., standard error) and indicate what it is in the legend.

Author Response

Response to Reviewer 2 Comments

Point 1: The authors describe a study which aimed to investigate the presence of ranavirus in various “ecological compartments” before and during a die-off. They also say that their aim is to provide some guidelines to improve ranavirus survey methods and allow more effective virus surveillance programmes to be implemented. I found the paper to be well-written and interesting in terms of the first objective, but have noted several points that could be clarified to make the manuscript stronger. The second objective (providing ranavirus survey recommendations for surveillance programs) could be strengthened by spending more time in the discussion making concrete recommendations.

Response 1: We agree with the last comments and make some adding in the relevant part of the manuscript (recommandation)

Point2: Title: Can you create a title that is more active and specific? Something that indicates what you found rather than the general topic would be valuable.

Response 2: We propose to adapt the title to the reviewer’ comment as:

eDNA increases the detectability of Ranavirus infection in an Alpine amphibian population

Point 3: Introduction: Lines 42-45: See new paper by Scheele et al. to update this information

Scheele, B. C., Pasmans, F., Skerratt, L. F., Berger, L., Martel, A., Beukema, W., ... & De la Riva, I. (2019). Amphibian fungal panzootic causes catastrophic and ongoing loss of biodiversity. Science, 363(6434), 1459-1463.

Response 3: Done (see comment by reviewer’ 1 who propose to shorten this part).

Point 4: Methods: Lines 79-86: I am confused about the description of the two lakes, as they were not sampled.

Response 4: We agree and delete this description. The paragraph is now:

The study was conducted in the Mercantour National Park in the southeastern Alps in France. The sampled area consisted of several small lakes and a pond. The Balaour pond (44.1082 N, 7.3742 E) is 25 x 20 m, and 2355 m a.s.l. It lies at a distance of about 100 m from the closest Lake. The maximum depth is 1.5 m and the bottom is granitic rock, partly covered by a shallow (max 0.10 m) sedimentary layer of mud. There is no macrophytic vegetation.

The common frog, Rana temporaria, uses these kinds of water bodies for breeding and hibernation in this alpine region [32]. In the study area, the common frog is the only amphibian species present, and breeds only in the Balaour pond (no breeding in the neighbouring lakes). Many lakes are stocked with brown trout (Salmo trutta) and the common minnow (Phoxinus phoxinus), used as bait, has also been introduced in the lakes. However, there are no trout or minnows in the studied pond

Point 5: Lines 104-118 are almost completely redundant with Table 1. Can you include some additional information (e.g., Gosner stage) to the table and remove the text?

Response 5: We agree and change the table 1 following also the reviewer 1 comments

Point 6: Lines 104-118: I assume that unique adult animals were sampled on different visits and that you could recognize this due to the toe clips, but it would be good to list this explicitly.

Response 6: Yes, we did not sample the same individuals 2 times. This is now explained line 124 (“The toe clip allowed checking that unique adult specimen were sampled along the studied period”).

Point 7: Line 143: This filter pore size is larger than what I have seen. How does it relate to the size of ranavirus particles? Or are you capturing animal skin cells that have ranavirus on them?

Response 7: The filter is not a simple filter (eg 0.20 or 0.22 microns) used for water analysis. It is an encapsulated filter of 0.45 micron. We change the sentence to” The 2-L water sample was homogenized by gently shaking the bag to ensure the eDNA was evenly mixed throughout the sample, and the whole 2-L water sample was then filtered directly in the field through a VigiDNA 0.45 μm filter (SPYGEN, Le Bourget du Lac, France) using a sterile 100-mL syringe”.

Point 8: Lines 214-215: Based on your results it seems like prevalence was the number of positive replicates rather than the number of positive samples.

Response 8: We agree, the prevalence is the number of positive replicates among the 12 PCR performed. The sentence Line 220 is change to “The results were expressed in term of prevalence (number of positive replicates/total number of replicates per sample) and mean …..

Point 9 Results: Did any of the negative controls come up positive? Was there any evidence of inhibition, especially in soil or water samples? (Your methods say that you tested for contamination and inhibition, but not what you found).

Response 9: This is the same question than reviewer’ 3 (point 9) and reviewer 1 (point 25).

We changed the sentence to “After DNA extraction, the samples were tested for inhibition using real-time amplification following the protocol described in Biggs et al., 2015. All the samples were found not inhibited."

Point 10: Lines 254-255: This line is a little misleading because ranavirus was not detected in the insect community, which I would consider one of the “compartments of the studied ecosystem”.

Response 10: We agree and change “all” by “several”

Point 11: Line 230: typo Most/mosu

Response 11: Done

Point 12: Lines 286-290: Why were individual replicates considered positive/negative when using water sampling, but replicates were pooled to the sample level for the animal samples and sediment samples? (That is, if you had 1 of 3 water filter replicates that was positive you’d report a prevalence of 33% but if you had 1 of 3 animal replicates positive you’d call the sample negative. And you report 1/3 replicates being positive in the sediment analysis even though by your description in the methods [lines 167-168] it seems like the status of this sample would be ‘equivocal’.).

Response 12: We do not agree with this comment, but we change the sentences because it was perhaps not clear.

The reviewer description of prevalence with water samples is wrong “That is, if you had 1 of 3 water filter replicates that was positive you’d report a prevalence of 33%”

For the water sample (one sample per session), 12 PCR were performed and the Rv detectability is the number of positive PCR/total number of PCR. This is described in 3.3 and we did not change it.

For the sediment, paragraph 3.4. we performed 3 replicates per sample at each occasion (described in method line 181). We rerun the sample and found a similar detection (1/3). Thus, following what is described line 170-175 for tissue, we consider that ranavirus was not detected in the sediment. We change the sentence as “All the qPCR replicates were negative for samples collected at session 1, 2, 4 and 6. At sessions 3 and 5, 1 qPCR among the 3 was positive for both samples and the two samples was rerun. Only 1 positive replicate was observed again, and these two sediment samples were thus considered also as negative.

We suppress the sentence about ranavirus in sediment in the discussion section (Line 326).

For the adults and tadpoles, the prevalence is the number of positive specimen / nb of specimen tested (eg as in table 2 Line 284). For each specimen (tissue sample), three PCR were performed (Line 169). Only specimen considered positive were used to estimate the prevalence.

Point 13 Discussion: Lines 341-342: See recent manuscript by Mosher et al that investigates tail samples and finds they perform favorably.

Mosher, B. A., Brand, A. B., Wiewel, A. N., Miller, D. A., Gray, M. J., Miller, D. L., & Grant, E. H. C. (2018). Estimating occurrence, prevalence, and detection of amphibian pathogens: insights from occupancy models. Journal of Wildlife Diseases.

Response 13: We change the sentence (see point 46 of reviewer 1) and added this new reference.

Point 14: Line 357: Typo – asymptotic should be asymptomatic

Response 14: Done

Point 15: I think that your manuscript needs to mention that ranavirus is generally detected imperfectly, wither due to sampling error (failing to capture DNA in certain samples – your pooling of water and sediment samples helps account for this), PCR inhibition, low quantity DNA, etc. Separating the detection probability from the occurrence of ranavirus requires modeling, but nested data like yours (multiple samples per site, multiple replicates per sample) are a natural fit for these occupancy models. I don’t believe that you need to re-run your analyses, but I do think it is prudent to recommend that readers consider the imperfect detectability in future studies. The Mosher et al. manuscript mentioned above introduces these ideas and, like you, concludes that there is great value in being able to sample for ranavirus using non-lethal methods outside the short window where die-offs occur.

Response 15: We agree and thank the reviewer to suggest this paper. We use it to rephrase the discussion and Conclusion.

Point 16: Conclusions section: Can you make any concrete recommendations to others studying ranavirus in the field? How many water, sediment, or animal samples should folks be collecting to be certain of detecting ranavirus when present at various time points? I assume you need more animals or water samples early on, but how many? What laboratory methods do you recommend? Did you have contamination or PCR inhibition problems? What other suggestions might you have?

Response 16: We agree on the value of these adding but our data allows only to make recommendation on the method to detect ranavirus (eg from tissue or water samples). We do not have data to make recommendation on lab processes, contamination, inhibition, etc. We try to follow the reviewer’ comments by proposing more applied recommendations in the Conclusion section.

Point 17: Figure 2: I suggest making the barplots larger in some way, and maintaining the same 4 categories (tadpole alive, tadpole dead, adult alive, and water sample) in each plot to help the reader. You can use an asterisk or something along those lines to indicate visits when certain categories were not detected/not sampled. Perhaps you can merge the barplot with the temperature data on a single plot with a second y axis? The inferences from the barplots seem to be more important than temperature differences to me, so those should be highlighted and easier to see.

Response 17: We added an asterisk along the lines to indicate visits when certain categories were not detected/not sampled.

We agree that the figure 2 give too much importance to the temperature curves and we changed it completely to better present the results on ranavirus prevalence.

Point 18: Table 2: In my opinion this table should be appended.

Response 18: We put this table in Appendix (Supplementary material)

Point 19: Figure 3: The x-axis labels make it seem as though negative quantities were observed for some categories. Can you move the x-axis labels down and renumber the y-axis to start at 0, rather than at 0.1? Also, for estimates that are generated from multiple samples, please include a measure of precision (e.g., standard error) and indicate what it is in the legend.

Response 19: The figure was completely changed, including SD.

Reviewer 3 Report

lines 37-39: this is likely an underestimate; consider adding "documented" as a qualifier, i.e., "...with some 200 species experiencing documented collapse..."

line 91: how can you be so sure that there are no fish in the pond? Is it really that clear and small?

line 107: in my copy at least, here and hereafter the percent sign (%) is showing up as a degree sign (°).

line 108: were these searches organized in any way? transects?

line 137: how did you choose these locations?

lines 144-145: how long were these samples at room temperature before extraction?

line 165: how did you determine that amplification curves were "similar".

line 177: change "Genomic" to "Genetic"

line 181: elaborate on "compared with the BLAST programme". default settings? which database(s)?

lines 199-200: elaborate on the test for PCR inhibition. Also, the results of this need to be reported in the Results section

line 220: what is the estimated population size of this population?

lines 221-222: what were these occurrences? amplexed pairs? egg masses?

line 225: replace "Less" with "Fewer"

line 230: typo

line 266: where is closing ")"?

line 279: typo

lines 292-293: put these results into figure 3

lines 301-302: not exactly, since you didn't satisfy Koch's postulates. soften the language here.

line 320: be consistent throughout with "Common frog" or "common frog".

line 341: you caution that results from clips or swabs must be analysed cautiously but you did not mention in the Discussion that you may have gotten false negatives in the qPCR assays due to imperfect detection in the sampling techniques

Author Response

Response to Reviewer 3 Comments

Point 1: lines 37-39: this is likely an underestimate; consider adding "documented" as a qualifier, i.e., "...with some 200 species experiencing documented collapse..."

Response 1: Change done according to the new paper (Scheele et al., 2019)

Point 2: line 91: how can you be so sure that there are no fish in the pond? Is it really that clear and small?

Response 2: Yes, small, clear, shallow, and a strong monitoring pressure which allows certifying that this pond is fish free.

Point 3: line 107: in my copy at least, here and hereafter the percent sign (%) is showing up as a degree sign (°).

Response 3: We check this and change “°” to a percent sign (%) (and alcohol to ethanol)

Point 4: line 108: were these searches organized in any way? transects?

Response 4: The searches were an exhaustive visual scanning of the pond. We add: “The pond was visually scan to detect frogs resting on the bottom”.

Point 5: line 137: how did you choose these locations?

Response 5: We add “water sample was collected at 20 locations equally spaced around the edge….”

Point 6: lines 144-145: how long were these samples at room temperature before extraction?

Response 6: We change the sentence to: “The filter was filled with 80 mL of CL1 conservative buffer (SPYGEN), and stored at room temperature before DNA extraction”.

Point 6: line 165: how did you determine that amplification curves were "similar".

Response 6: It is explain in the brackets (shape, cycle threshold, values and > 0.1 genomic equivalent).

Point 7: line 177: change "Genomic" to "Genetic"

Response 7: There is no “Genomic” line 177…

Point 8: line 181: elaborate on "compared with the BLAST programme". default settings? which database(s)?

Response 8: new sentence “….and the sequences were compared with the BLAST program (default settings), to Ranavirus sequences previously identified in this region..

Point 9: lines 199-200: elaborate on the test for PCR inhibition. Also, the results of this need to be reported in the Results section

Response 9: new sentence “After DNA extraction, the samples were tested for inhibition using real-time amplification following the protocol described in Biggs et al., 2015. All the samples were found not inhibited.”

Point 10: line 220: what is the estimated population size of this population?

Response 10: The sentence is change to “The breeding population in the Balaour pond has long been observed, but the population size was not precisely estimated. The number of egg-masses are regularly around several hundred (M.-F. Lecchia, pers.comm.).

Point 11: lines 221-222: what were these occurrences? amplexed pairs? egg masses?

Response 11: “occurrences” changes to “egg-masses”

Point 12: line 225: replace "Less" with "Fewer"

Response 12: Done

Point 13: line 230: typo

Response 13: Done

Point 14: line 266: where is closing ")"?

Response 14: Done: after and [32]) of…….

Point 15: line 279: typo

Response 15: Done

Point 16: lines 292-293: put these results into figure 3.

Response 16: There is a risk of contamination of the sediment samples by water (explain line 314). The negative result (no ranavirus on session 1, 2 and 6, while detected in water) just indicate that this substrate did not play a major role for contamination. We consider that it is better to not represent “sediment” on fig. 3 because it will show 2 positive values (possibly contaminated) and 4 zero values.

Point 17: lines 301-302: not exactly, since you didn't satisfy Koch's postulates. soften the language here.

Response 17: We agree and change the sentence to: Ranavirus was identified as the etiologic agent of the mass mortality of common frog in south-eastern Alps [15]. Our findings in the monitored pond are in line with this result.

Point 18: line 320: be consistent throughout with "Common frog" or "common frog".

Response 18: We check the whole manuscript and keep only “common frog”

Point 19: line 341: you caution that results from clips or swabs must be analysed cautiously but you did not mention in the Discussion that you may have gotten false negatives in the qPCR assays due to imperfect detection in the sampling techniques.

Response 19: We agree for this comment. Our results show a rather good detection because the quantity of Rv+ frogs was 3 and 4 among the 5 tested in June and July. On 10 June and 19 September, the 5 tested frogs were Rv-.We consider that this recommendation (be careful with data collected only with toes clip or swabs) is important and has to stay where it it (line 343). 

Reviewer 4 Report

The authors present a sound and informative study on the application of eDNA for the detection of EIDs in the natural system.I feel the paper is well written, well presented and highly informative. I have two minor questions:

1. Are there really no other amphibian species in the study ponds?

2. Is it possible to make a table combining table 1 and 2? It’d be interesting to see when the various samples tested, either positive or negative(insects, sediment and animals) over time. All information is in the text, but an overview would be convenient for the reader.

Author Response

Response to Reviewer 4 Comments

Point 1: 1. Are there really no other amphibian species in the study ponds?

Response 1: YES. The sentence “In the study area, the common frog is the only amphibian species present….“ is added line 88.

Point 2: 2. Is it possible to make a table combining table 1 and 2? It’d be interesting to see when the various samples tested, either positive or negative (insects, sediment and animals) over time. All information is in the text, but an overview would be convenient for the reader

Response 2: This is done, also in the new figure 2

Round 2

Reviewer 1 Report

I appreciate the authors' interest in improving the manuscript and I think it is becoming much easier to follow and pull out interesting information. (And I remain convinced that by studying multiple components of the system through time, this study helps us understand how ranavirus epidemics unfold.)  Still, while the author's made most of the specific suggestions, the broader issue, that the manuscript lacks a clear point of view and organization around this point of view, remains.  I think this could be addressed and would make this manuscript much stronger and I have provided more concrete suggestions for how they might re-organize their introduction and discussion. They are, of course, welcome to ignore my suggestions, but they need to provide a much clearer and more clearly developed sense of what this manuscript adds to the larger discussion of ranaviruses or disease ecology.

The introduction was somewhat better, but its organization was still uncertain, which again left me wondering what the authors were trying to determine and why. The argument seems to be that amphibians are declining, pathogens are one of the reasons, so the authors are going to describe some particular aspects of this very particular system. What is missing for me is the connection between the big issues and the particular study. That is, the introduction does not set up this particular project very well.

For instance, the authors say they want to describe infection status through time, but they have not explained why this is important or interesting or what question(s) it will help answer. I happen to think it is inherently interesting, but that is because I would like to know what causes epidemics to become die-offs, where the infections are coming from or which routes of infection are most important, and why epidemics seem to unfold slowly when ranavirus is so easily transmitted in the laboratory. That is, there are open questions that I think this sort of study can help address. I would find this paper much, much more interesting if those questions were set out clearly rather than sort of implied. 

Similarly, the second aim is about comparing sampling methods, and this topic is introduced somewhat better in the introduction, but it is not clear what question(s) need to be answered that have not been already answered. That is, what does this study provide that, for instance, Hall et al's study did not? (Hall et al. [2016 & 2018] collected eDNA two or several times in a season, respectively, and compared them with virus from animal tissues.) There was somewhat more of a defense in the response to reviewer comments, though I had trouble following it, so I think they have a sense of this. I might suggest that extending the comparison between eDNA and samples of animals to other systems is important and that adding the sediment and insects compartments helps complete the whole picture.  In any case, I would suggest they be very clear about what they bring to the discussion, how they will improve surveillance or our understanding of how eDNA-based surveillance works. Lastly, and relatedly, they state they will  provide guidelines for better implementation of virus surveillance programs, but I found the last paragraph of the paper did not really offer any concrete guidelines and essentially said that people should do what most people are already doing... if they are not going to set out clear guidelines with clear rationale, they should remove this promise.

I found figure 2 to be much more compelling and I liked the pictures of the stages for reference. I spent quite a bit of time trying to understand all it had to say. However, there were a few issue that made it hard to read. First, the y-axis for the Rv quantities varied between sub-graphs or panels or whatever we should call them. This made it very difficult to compare virus quantities between sampling periods. Please use a consistent axis. The second minor issue was that the text on these y-axis labels were so small it was hard to read. Third, you appear to give the water samples a sample size of 12, but these are not 12 water samples, but the 12 wells of qPCR reactions for a single homogenized water sample. This is not at all the same as sample size of 5 tadpoles or 5 adults, which are independent of each other.  I think you should just remove these bars for "prevalence" of the water samples. In fact, I think you could probably remove all of the prevalence bars and instead write somewhere "1/5", "1/1", and so on. But I leave that up to you. Lastly, I found myself spending a great deal of time looking between the bars and the legend to remind myself what each color represented. My preferences would be one horizontal panel or facet for each "compartment" (e.g., one for the Rv quantities in tadpoles, one for adults, one for water... you could have a separate symbol for the dead individuals), but if you prefer using bars, perhaps you could use labels underneath each panel (e.g., "T(a)", "T(d)", "A(a)", "A(d)", "W") to make it easier to track. 

The discussion still jumps from one topic to another and I had a very hard time following the authors' logic. Consider the first paragraph. It starts with a statement about ranavirus being the etiological agent of a die-off in another pond not in this study, which seems a very strange place to start, then moves on to saying all of the dead tadpoles had ranavirus. Then we read about egg masses, the size of the breeding population, and the synchrony of breeding. And finally we end with breeding aggregations potentially facilitating transmission, but only one dead frog being found. The _next_ paragraph continues with infected adults being found in July, moves onto ranavirus in another pond, talks about breeding again, moves on to ranavirus not being found insects or sediments, and then ends on the question of persistence between years. Neither paragraph seems like a cohesive whole and neither really develops any of the ideas presented very fully.  The rest of the discussion seemed similarly disjointed. It needs a much clearer organization with one  main idea per paragraph and a clearly developed ideas.

I would suggest the authors organize their discussion into the following topics:

1) The utility of the study: it is one of very few that documents the dynamics in a pond through time, and unique in that it considers different life stages and abiotic components (but of course, Hall et al. 2018 used eDNA, too). This provides a comprehensive view of the dynamics.

2) The dynamics of the Rv epidemic in tadpoles, from rare, low-level infections to common, but low-level infections, to a die-off (and I cannot really see what happens with the live tadpoles), to moderately intense infections for several weeks, and then seeming to become less common. (I would be cautious in interpreting the prevalence too strongly as your sample size is only n=5) 

2b) I think you can add in here or in a separate paragraph the eDNA information. But you should try to interpret what it means. I might suggest is means that infected hosts are shedding virus into the water, but the amounts are rather low and do not seem to change much (as far as I can tell). This, to me, raises the question of what causes the die-off if it is not greater exposure to ranavirus (at least not in the water). 

3) The potential causes of the die-off. This includes the stage and temperature hypotheses already discussed.

4) The role of the adults in the epidemiology. Did they introduce the virus? Did breeding aggregations potentially facilitate transmission? Are they important in persistence?

5) Maybe talk about persistence separately here... I would also bring in the lack of detections in insects and sediments here.

6) The potential advantages and disadvantages of eDNA sampling. This would include that water sample was positive before the die-off was observed, when Rv intensities in tadpoles were in the hundreds of copies, and remained positive in the samples after the die-off. But it also was negative the previous two periods when ranavirus was present, but at very low levels. All of which suggests that eDNA samples can detect moderate to high-levels of infection, as in the wood frog system, but may be poor at detecting very low-level or rare infections, like those you observed early in the season. eDNA-based sampling for ranavirus might want to focus on mid to late season time points (see also Julian et al. 2019).

Lastly, I would ask the authors to take more care in their submitted manuscripts. I can very much appreciate the difficult of writing a manuscript in another language, but this manuscript was full of simple errors that are easily seen. "Ranavirus" was misspelled, sentences seemed to begin with periods, and there were paragraphs with only one sentence making me wonder whether someone hit the "return" key on the computer without meaning to do so.

Specific comments

L51-54: The logic here is confusing. You seem to be saying that because ranavirus can be transmitted between taxa you need to sample all of these taxa. I do not know if that is generally true---I would guess tadpoles are _much_ more likely to be infected from other tadpoles than from snakes or turtles or fish---and _you_ do not sample these other taxa in this study. Lastly, are you interested in "individual" risk of infection or something else, like the dynamics of ranavirus in a population?

L68-9: I know you replaced "compartments" with "hosts," but sediment and water are not hosts. 

L70-71: If you are going to compare sample types (tissues from hosts to eDNA from water), you need to do so formally, which it seems you do not do. 

L101: I think "plastic bottle" may have been a centrifuge tube or snap-cap tube. Is that right?

L103: Please clarify that the distal phalange was taken from _adults_.

L111/Table 1: I do not think this table provides enough information to justify it being here. Fig. 2 and the M&M provide all of the details already. 

L158-9: Please state that this is a Taqman realtime quantitative PCR assay. It was not clear from context. Second, please provide the authors rather than just the reference number (i.e., "Leung et al.") 

L160: if this was quantitative, and I think it was, what was used as the quantitative standard?

L191-2: Please add a phrase such as, "which involved adding a synthetic DNA sequence to each sample and then trying to amplify it" to the end of the sentence so that the reader does not have to read another paper to have a sense of how this was done. 

L196-198: Here it seems you provide the sequences of Leung et al's qPCR assay, but you do not reference it. I think you can clarify that it is the same assay as above, provide the reference, and not include the sequence. Also, were the reaction conditions the same as in reference 15? If so, you can leave them out. Indeed, if they are the same or very similar, you could have one qPCR section rather than one very short and one very long sections as you have now. If they are different, please explain how they differed. As written it makes it seem like these were entirely different assays done differently. 

L206-8: This is not actually a measure of prevalence and I do not think it provides any information that the copy number does not provide already. I would strongly suggest removing this and references to "prevalence" of the water samples throughout to avoid confusion.  

L211-212: I am confused. The qPCR reaction of Leung et al. should amplify a 97bp region, but here you say you sequenced a 530 bp region. I am guessing there was a secondary conventional PCR reaction with something like Mao et al. MCP4/5 primers to amplify this larger region, w which was then sequenced. Is this correct? In any case, it needs to be clarified.

References:

Hall, E. M., C. S. Goldberg, J. L. Brunner, and E. J. Crespi. 2018. Seasonal dynamics and potential drivers of ranavirus epidemics in wood frog populations. Oecologia 188:1253-1262. 

Julian, J. T., G. W. Glenney, and C. Rees. 2019. Evaluating observer bias and seasonal detection rates in amphibian pathogen eDNA collections by citizen scientists. Diseases of Aquatic Organisms 134:15-24. 

Author Response

Response to reviewer’ 1

Q1 : I appreciate the authors' interest in improving the manuscript and I think it is becoming much easier to follow and pull out interesting information. (And I remain convinced that by studying multiple components of the system through time, this study helps us understand how ranavirus epidemics unfold.)  Still, while the author's made most of the specific suggestions, the broader issue, that the manuscript lacks a clear point of view and organization around this point of view, remains.  I think this could be addressed and would make this manuscript much stronger and I have provided more concrete suggestions for how they might re-organize their introduction and discussion. They are, of course, welcome to ignore my suggestions, but they need to provide a much clearer and more clearly developed sense of what this manuscript adds to the larger discussion of ranaviruses or disease ecology.

R1: Again we thank the reviewer’ investment to improve the manuscript and try to follow the recommendations.

Q2: The introduction was somewhat better, but its organization was still uncertain, which again left me wondering what the authors were trying to determine and why. The argument seems to be that amphibians are declining, pathogens are one of the reasons, so the authors are going to describe some particular aspects of this very particular system. What is missing for me is the connection between the big issues and the particular study. That is, the introduction does not set up this particular project very well.

R2: We could start directly the paper with the main goal, i.e. the description of the use of eDNA to detect the virus in an alpine lake. But we consider that it is always interesting to start with a more global view of the context, and it is why we started with some data on amphibian “at risk” and especially disease. The connexion between this global issue (amphibian disease) and our particular study is done by the sentence “Rapid and accurate identification of pathogenic microorganisms is essential for the early detection of infection and the deployment of appropriate mitigation measures (Line 47-48).

Q3: For instance, the authors say they want to describe infection status through time, but they have not explained why this is important or interesting or what question(s) it will help answer. I happen to think it is inherently interesting, but that is because I would like to know what causes epidemics to become die-offs, where the infections are coming from or which routes of infection are most important, and why epidemics seem to unfold slowly when ranavirus is so easily transmitted in the laboratory. That is, there are open questions that I think this sort of study can help address. I would find this paper much, much more interesting if those questions were set out clearly rather than sort of implied. 

R3: We agree that these questions (what causes epidemics to become die-offs, where the infections are coming from or which routes of infection are most important, etc.) are fundamental, but it is not the aim of this “Technical note”. It is clearly a technical approach (see next Question). We try to add some results (e.g. sampling in other compartments than amphibians) to improve the knowledge on this pathogen and the methods to detect it. We assume this (“Technical note”) in rephrasing the end of the introduction (objectives) and with the new orientation of the Discussion (which clearly focus on “technics”).

Q4: Similarly, the second aim is about comparing sampling methods, and this topic is introduced somewhat better in the introduction, but it is not clear what question(s) need to be answered that have not been already answered. That is, what does this study provide that, for instance, Hall et al's study did not? (Hall et al. [2016 & 2018] collected eDNA two or several times in a season, respectively, and compared them with virus from animal tissues.) There was somewhat more of a defense in the response to reviewer comments, though I had trouble following it, so I think they have a sense of this. I might suggest that extending the comparison between eDNA and samples of animals to other systems is important and that adding the sediment and insects compartments helps complete the whole picture.  In any case, I would suggest they be very clear about what they bring to the discussion, how they will improve surveillance or our understanding of how eDNA-based surveillance works.

R4: We agree and rewrite this paragraph in order to better focus on how eDNA can improve pathogen detection. We clearly describe Hall et al 2016 results in the Introduction and are back to it Hall et al 2018 in the discussion (new paragraph on how eDNA can improve pathogen detection).

We rephrase the last sentences of the Introduction in order to better match with what will be provided in the paper; and limit the recommendations on some guidelines for the use or eDNA (and occupancy modelling, according to reviewer’s 2 comments).

Q5 Lastly, and relatedly, they state they will  provide guidelines for better implementation of virus surveillance programs, but I found the last paragraph of the paper did not really offer any concrete guidelines and essentially said that people should do what most people are already doing... if they are not going to set out clear guidelines with clear rationale, they should remove this promise.

R5: We agree and according to the R4, we added a paragraph in the discussion and change the last sentence in the Introduction.

Line 64-72: “This study had two key aims: (1) to describe the infection status of the different potential hosts of ranavirus (adult frogs, tadpoles, insects) and ecosystem compartments (sediment and water) in an alpine ecosystem of the common frog during its main activity period (summer), both prior to and after an observed die-off, and (2) to use the eDNA method to detect Ranavirus along this infection event. By comparing previous studies using eDNA to detect Chytrids and Ranavirus, we provide some recommendations for a better implementation of water sampling in e.g. surveillance programmes.”

Q6: I found figure 2 to be much more compelling and I liked the pictures of the stages for reference. I spent quite a bit of time trying to understand all it had to say. However, there were a few issue that made it hard to read. First, the y-axis for the Rv quantities varied between sub-graphs or panels or whatever we should call them. This made it very difficult to compare virus quantities between sampling periods. Please use a consistent axis.

R6: We provide a new draft of Fig. 2 following these comments:

y-axis (Rv quantities). We use a consistent axis for all the dates except for July where the Rv quantities are very high (> 107). We also now use a logarithmic axis.. We hope that this solution will be agreed.

Q7: The second minor issue was that the text on these y-axis labels were so small it was hard to read.

R7: We increase the text size.

Q8: Third, you appear to give the water samples a sample size of 12, but these are not 12 water samples, but the 12 wells of qPCR reactions for a single homogenized water sample. This is not at all the same as sample size of 5 tadpoles or 5 adults, which are independent of each other.  I think you should just remove these bars for "prevalence" of the water samples. In fact, I think you could probably remove all of the prevalence bars and instead write somewhere "1/5", "1/1", and so on. But I leave that up to you.

R8: We agree with this comment and suppress the “prevalence” data for water samples.

Q9: Lastly, I found myself spending a great deal of time looking between the bars and the legend to remind myself what each color represented. My preferences would be one horizontal panel or facet for each "compartment" (e.g., one for the Rv quantities in tadpoles, one for adults, one for water... you could have a separate symbol for the dead individuals), but if you prefer using bars, perhaps you could use labels underneath each panel (e.g., "T(a)", "T(d)", "A(a)", "A(d)", "W") to make it easier to track.

R9: We prefer to keep the figure in its current design. There is one colour per category and only 5 categories. Moreover, each categories is always at the same place in the sub-graphs. We could add some labels as proposed by the reviewer above each bar, but we consider that it is repetitive to the currently available legend.

The discussion still jumps from one topic to another and I had a very hard time following the authors' logic. Consider the first paragraph. It starts with a statement about ranavirus being the etiological agent of a die-off in another pond not in this study, which seems a very strange place to start, then moves on to saying all of the dead tadpoles had ranavirus. Then we read about egg masses, the size of the breeding population, and the synchrony of breeding. And finally we end with breeding aggregations potentially facilitating transmission, but only one dead frog being found. The _next_ paragraph continues with infected adults being found in July, moves onto ranavirus in another pond, talks about breeding again, moves on to ranavirus not being found insects or sediments, and then ends on the question of persistence between years. Neither paragraph seems like a cohesive whole and neither really develops any of the ideas presented very fully.  The rest of the discussion seemed similarly disjointed. It needs a much clearer organization with one  main idea per paragraph and a clearly developed ideas.

I would suggest the authors organize their discussion into the following topics:

Q10: 1) The utility of the study: it is one of very few that documents the dynamics in a pond through time, and unique in that it considers different life stages and abiotic components (but of course, Hall et al. 2018 used eDNA, too). This provides a comprehensive view of the dynamics.

2) The dynamics of the Rv epidemic in tadpoles, from rare, low-level infections to common, but low-level infections, to a die-off (and I cannot really see what happens with the live tadpoles), to moderately intense infections for several weeks, and then seeming to become less common. (I would be cautious in interpreting the prevalence too strongly as your sample size is only n=5)

We agree and started the discussion with the description of the dynamics in a pond through time, also closely referring to Hall et al. 2018.

2b) I think you can add in here or in a separate paragraph the eDNA information. But you should try to interpret what it means. I might suggest is means that infected hosts are shedding virus into the water, but the amounts are rather low and do not seem to change much (as far as I can tell). This, to me, raises the question of what causes the die-off if it is not greater exposure to ranavirus (at least not in the water).

This part is now in the new paragraph on eDNA

3) The potential causes of the die-off. This includes the stage and temperature hypotheses already discussed.

4) The role of the adults in the epidemiology. Did they introduce the virus? Did breeding aggregations potentially facilitate transmission? Are they important in persistence?

5) Maybe talk about persistence separately here... I would also bring in the lack of detections in insects and sediments here.

These 3 points are in the new discussion section.

6) The potential advantages and disadvantages of eDNA sampling. This would include that water sample was positive before the die-off was observed, when Rv intensities in tadpoles were in the hundreds of copies, and remained positive in the samples after the die-off. But it also was negative the previous two periods when ranavirus was present, but at very low levels. All of which suggests that eDNA samples can detect moderate to high-levels of infection, as in the wood frog system, but may be poor at detecting very low-level or rare infections, like those you observed early in the season. eDNA-based sampling for ranavirus might want to focus on mid to late season time points (see also Julian et al. 2019).

We agree and produce a completely new paragraph on eDNA for pathogen detection (with a new table).

Q11 : Lastly, I would ask the authors to take more care in their submitted manuscripts. I can very much appreciate the difficult of writing a manuscript in another language, but this manuscript was full of simple errors that are easily seen. "Ranavirus" was misspelled, sentences seemed to begin with periods, and there were paragraphs with only one sentence making me wonder whether someone hit the "return" key on the computer without meaning to do so.

R11: We solved the remaining mistakes (Ranavirus, periods, paragraphs…).

Specific comments

L51-54: The logic here is confusing. You seem to be saying that because ranavirus can be transmitted between taxa you need to sample all of these taxa. I do not know if that is generally true---I would guess tadpoles are _much_ more likely to be infected from other tadpoles than from snakes or turtles or fish---and _you_ do not sample these other taxa in this study. Lastly, are you interested in "individual" risk of infection or something else, like the dynamics of ranavirus in a population?

Frogs (and tadpoles) are the only potential vertebrate host in this pond. We agree that these sentences are not relevant (the multi host aspect is presented before, Line 41). We are not looking at individual risk…

L68-9: I know you replaced "compartments" with "hosts," but sediment and water are not hosts.

We agree and adapted the sentence to “….to describe the infection status of different potential hosts of Ranavirus (common frog adults, tadpoles, insects) and ecosystem compartments (sediment and water) in an alpine lake during the activity period.

L70-71: If you are going to compare sample types (tissues from hosts to eDNA from water), you need to do so formally, which it seems you do not do. 

This comparison is provided Line 289: “However, the GE values in infected live tadpoles and the number of DNA Ranavirus copies in water were not significantly correlated (Spearman rank correlation R = 0.25, p>0.05). “

L101: I think "plastic bottle" may have been a centrifuge tube or snap-cap tube. Is that right?

We change “bottle” with “snap-cap tube

L103: Please clarify that the distal phalange was taken from _adults_.

We suppress the coma between the two sentences.

L111/Table 1: I do not think this table provides enough information to justify it being here. Fig. 2 and the M&M provide all of the details already.

Well….This table was asked in a former round….We keep it and ask the opinion of the editor for leaving or suppressing it.

L158-9: Please state that this is a Taqman realtime quantitative PCR assay. It was not clear from context.

Done

Second, please provide the authors rather than just the reference number (i.e., "Leung et al.")

Done

L160: if this was quantitative, and I think it was, what was used as the quantitative standard?

Done : “Ranavirus DNA, provided in several densities by Stephen Price (Zoological Society, London, UK), was used for the standard curve.”

L191-2: Please add a phrase such as, "which involved adding a synthetic DNA sequence to each sample and then trying to amplify it" to the end of the sentence so that the reader does not have to read another paper to have a sense of how this was done. 

Done

L196-198: Here it seems you provide the sequences of Leung et al's qPCR assay, but you do not reference it. I think you can clarify that it is the same assay as above, provide the reference, and not include the sequence. Also, were the reaction conditions the same as in reference 15? If so, you can leave them out. Indeed, if they are the same or very similar, you could have one qPCR section rather than one very short and one very long sections as you have now. If they are different, please explain how they differed. As written it makes it seem like these were entirely different assays done differently.

We added the reference to Leung et al (2017), and suppress the sequences. The reaction conditions are different from the conditions described in ref [13] (former 15) and are precisely described in this paragraph.

L206-8: This is not actually a measure of prevalence and I do not think it provides any information that the copy number does not provide already. I would strongly suggest removing this and references to "prevalence" of the water samples throughout to avoid confusion.

We agree and suppressed the term “prevalence” for water samples (it was also suppressed in fig. 2)

L211-212: I am confused. The qPCR reaction of Leung et al. should amplify a 97bp region, but here you say you sequenced a 530 bp region. I am guessing there was a secondary conventional PCR reaction with something like Mao et al. MCP4/5 primers to amplify this larger region, which was then sequenced. Is this correct? In any case, it needs to be clarified.

We agree that this secondary PCR reaction was performed according to the protocol described in Mao et al. (1999). This is corrected in text.

References:

Hall, E. M., C. S. Goldberg, J. L. Brunner, and E. J. Crespi. 2018. Seasonal dynamics and potential drivers of ranavirus epidemics in wood frog populations. Oecologia 188:1253-1262.

Added 

Julian, J. T., G. W. Glenney, and C. Rees. 2019. Evaluating observer bias and seasonal detection rates in amphibian pathogen eDNA collections by citizen scientists. Diseases of Aquatic Organisms 134:15-24.

Added

ABSTRACT: We trained volunteers from conservation organizations to collect environmental DNA (eDNA) from 21 ponds with amphibian communities that had a history of Batrachochytrium dendrobatidis (Bd) and ranavirus (Rv) infections. Volunteers were given sampling kits to filter pond water and preserve eDNA on filter paper, as were the principal investigators (PIs), who made independent collections within 48 h of volunteer collections. Using multi-scale occupancy modeling, we found no evidence to suggest the observer who collected the water sample (volunteer or PI) influenced either the probability of capturing eDNA on a filter or the probability of detecting extracted eDNA in a quantitative PCR (qPCR) reaction. The cumulative detection probability of Bd eDNA at a pond decreased from May through July 2017 because there was a decrease in the probability of detecting eDNA in qPCR reactions. In contrast, cumulative detection probability increased from May to July for Rv due to a higher probability of capturing eDNA on filters later in the year. Our models estimate that both pathogens could be detected with 95% confidence in as few as 5 water samples taken in June or July tested with either 4 or 3 qPCR reactions, respectively. Our eDNA protocols appeared to detect pathogens with 95% confidence using considerably fewer samples than protocols which typically recommend sampling ≥30 individual animals. In addition, eDNA sampling could reduce some biosecurity concerns, jurisdictional and institutional permitting, and stress to biota at ponds.

Brittany A. Mosher1 · Kathryn P. Huyvaert1 · Larissa L. Bailey1

Understanding the ecosystem-level persistence of pathogens is essential for predicting and measuring host–pathogen dynamics.

However, this process is often masked, in part due to a reliance on host-based pathogen detection methods. The amphibian

pathogens Batrachochytrium dendrobatidis (Bd) and B. salamandrivorans (Bsal) are pathogens of global conservation concern.

Despite having free-living life stages, little is known about the distribution and persistence of these pathogens outside

of their amphibian hosts. We combine historic amphibian monitoring data with contemporary host- and environment-based

pathogen detection data to obtain estimates of Bd occurrence independent of amphibian host distributions. We also evaluate

differences in filter- and swab-based detection probability and assess inferential differences arising from using different

decision criteria used to classify samples as positive or negative. Water filtration-based detection probabilities were lower

than those from swabs but were > 10%, and swab-based detection probabilities varied seasonally, declining in the early fall.

The decision criterion used to classify samples as positive or negative was important; using a more liberal criterion yielded

higher estimates of Bd occurrence than when a conservative criterion was used. Different covariates were important when

using the liberal or conservative criterion in modeling Bd detection. We found evidence of long-term Bd persistence for

several years after an amphibian host species of conservation concern, the boreal toad (Anaxyrus boreas boreas), was last

detected. Our work provides evidence of long-term Bd persistence in the ecosystem, and underscores the importance of

environmental samples for understanding and mitigating disease-related threats to amphibian biodiversity.

Accurate pathogen detection is essential for developing management strategies to address emerging infectious diseases, an increasingly prominent threat to wildlife. Sampling for freeliving pathogens outside of their hosts has benefits for inference and study efficiency, but is still uncommon. We used a laboratory experiment to evaluate the influences of pathogen concentration, water type, and qPCR inhibitors on the detection and quantification of Batrachochytrium dendrobatidis (Bd) using water filtration. We compared results pre and postinhibitor removal, and assessed inferential differences when single versus multiple samples were collected across space or time. We found that qPCR inhibition influenced both Bd detection and quantification in natural water samples, resulting in biased inferences about Bd occurrence and abundance. Biases in occurrence could be mitigated by collecting multiple samples in space or time, but biases in Bd quantification were persistent. Differences in Bd concentration resulted in variation in detection probability, indicating that occupancy modeling could be used to explore factors influencing heterogeneity in Bd abundance among samples, sites, or over time. Our work will influence the design of studies involving amphibian disease dynamics and studies utilizing environmental DNA (eDNA) to understand species distributions.

Reviewer 2 Report

The authors have done a nice job of reworking the manuscript and only a few small issues remain.

1 - Figure 3 is referenced (5 times) in the text but it appears that it was added to Figure 2 and there is now no Figure 3. 

2 - Many of the papers you cite (Chestnut et al., Schmidt et al., both Mosher et al.) explicitly account for imperfect detectability in pathogen detection methods. In my last review I suggested adding some mention of this topic to the discussion, but it doesn't look like that has been done. I think it needs to be mentioned that the estimates of prevalence shown here are likely underestimates (and technically should be called 'naive' prevalence) and that methods exist for accounting for imperfect detection (i.e., occupancy models).

3-  In the literature cited the formatting is inconsistent. Sometimes manuscript title capitalization is inconsistent (for example, see entries for citations 18 and 19). Citation 17's author is written as "Anonymous". Citation 70 is redundant with citation 30 (I believe citation 70 should be the Mosher et al. Journal of Wildlife Diseases paper I suggested last time). These are just examples that I noticed with a quick glance - please take care to edit the literature cited section.

Author Response

Response to reviewer’ 2

The authors have done a nice job of reworking the manuscript and only a few small issues remain.

We thank the reviewers for their very constructive comments

1 - Figure 3 is referenced (5 times) in the text but it appears that it was added to Figure 2 and there is now no Figure 3.

We check the manuscript and suppress Fig. 3.

2 - Many of the papers you cite (Chestnut et al., Schmidt et al., both Mosher et al.) explicitly account for imperfect detectability in pathogen detection methods. In my last review I suggested adding some mention of this topic to the discussion, but it doesn't look like that has been done. I think it needs to be mentioned that the estimates of prevalence shown here are likely underestimates (and technically should be called 'naive' prevalence) and that methods exist for accounting for imperfect detection (i.e., occupancy models).

We fully agree and added a specif paragraph on this aspect (+ table 2). It is clear that the use of eDNA has to be – when possible – implemented with an occupancy design…

3-  In the literature cited the formatting is inconsistent. Sometimes manuscript title capitalization is inconsistent (for example, see entries for citations 18 and 19). Citation 17's author is written as "Anonymous". Citation 70 is redundant with citation 30 (I believe citation 70 should be the Mosher et al. Journal of Wildlife Diseases paper I suggested last time). These are just examples that I noticed with a quick glance - please take care to edit the literature cited section.

We check the literature to solve several problems…
